# Fusion Surface Models: 2+1d Lattice Models from Fusion 2-Categories

Kansei Inamura[1] and Kantaro Ohmori[2]

[1]Institute for Solid State Physics, University of Tokyo, Kashiwa, Chiba 277-8581, Japan
[2]Department of Physics, The University of Tokyo, Bunkyo-ku, Tokyo 113-0033, Japan

**Abstract**

We construct (2+1)-dimensional lattice systems, which we call fusion surface models. These models have finite non-invertible symmetries described by general fusion 2-categories. Our method can be applied to build microscopic models with, for example, anomalous or non-anomalous one-form symmetries, 2-group symmetries, or non-invertible one-form symmetries that capture non-abelian anyon statistics. The construction of these models generalizes the construction of the 1+1d anyon chains formalized by Aasen, Fendley, and Mong. Along with the fusion surface models, we also obtain the corresponding three-dimensional classical statistical models, which are 3d analogues of the 2d Aasen-Fendley-Mong height models. In the construction, the "symmetry TFTs" for fusion 2-category symmetries play an important role.

# 1 Introduction and Summary

## 1.1 Motivation

Symmetry plays a fundamental role in both constructing and analyzing models of physical systems. Recently, various generalizations of symmetry have been introduced, including higher-form symmetry [1], higher-group symmetry [2–5], and even more general non-invertible symmetry [6,7]. These generalized symmetries greatly extend the applicability of various symmetry-based techniques in theoretical physics, and have therefore been one of the main topics in the field.

The core principle of the generalizations is the correspondence between symmetry operations and *topological defects/operators* [1], see Figure 1 for an illustration of this correspondence. In particular, symmetry operations for a conventional symmetry correspond to invertible topological defects with codimension one, which form a group under the fusion. The generalizations of a conventional symmetry are achieved by relaxing the requirements for the dimensionality and invertibility of topological defects: higher-form symmetries are generated by topological defects with higher codimensions and non-invertible symmetries

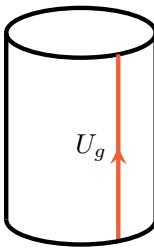
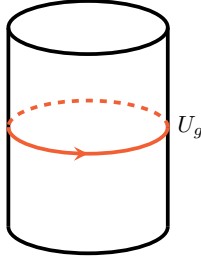

Twisted boundary condition

Symmetry action on the state space

Figure 1: A timelike insertion of a symmetry defect realizes a twisted boundary condition, while a spacelike insertion does a symmetry operation on a state.

are generated by topological defects that do not have their inverses.[1] We note that topological defects associated with a non-invertible symmetry can have arbitrary codimensions, and therefore non-invertible symmetries include higher-form symmetries as special cases. While higher-form symmetries are still described by groups, non-invertible symmetries are no longer described by groups in general because the fusion rules of the associated topological defects are not necessarily group-like.

In 1+1 dimensions, finite non-invertible symmetries are generally described by fusion categories [6,7,9] (see also [10] for an earlier reference), which are natural generalizations of finite groups.[2] For this reason, finite non-invertible symmetries in 1+1 dimensions are called fusion category symmetries [19]. Fusion category symmetries are particularly well studied in the context of rational conformal field theories [7,20–26] and topological field theories (TFTs) [6,10,19,27–30]. See also, e.g., [31–52] for recent developments. Although these symmetries were originally discussed in the context of quantum field theories (QFTs), they also exist on the lattice. In particular, we can systematically construct 1+1d lattice models with general fusion category symmetries, which are known as anyon chain models [33,53,54].

In higher dimensions, finite non-invertible symmetries in univery theories are expected to be described by fusion higher categories [55–58][3]. Since the discovery of concrete realizations of such symmetries in lattice models [59] and QFTs [60–64], non-invertible symmetries in higher dimensions have been studied intensively in various contexts, see, e.g., [59–121] for recent advances and also [122, 123] for earlier discussions. However, systematic construction of physical systems with general fusion higher category symmetries is still lacking.

Given the generalization of symmetry, one may wonder whether we can utilize it to build physical models with a given generalized symmetry. This question was answered affirmatively by Aasen, Fendley, and Mong for fusion category symmetries in 1+1 dimen-

---

[1]One can further generalize the notion of symmetry by allowing the defects to be non-topological along some spatial directions. Symmetries generated by such defects are called subsystem symmetries, which are typically exhibited by fractonic systems [8]. We do not investigate this direction in this paper.

[2]Precisely, while finite non-invertible symmetries of 1+1d bosonic systems are described by fusion categories, finite non-invertible symmetries of 1+1d fermionic systems are described by superfusion categories [11–18]. In this paper, we will only consider bosonic systems. Technically, all fusion categories and fusion 2-categories that we will discuss in this paper are supposed to be spherical. We also emphasize that we do not consider non-unitary theories where symmetry categories can be non-semisimple.

[3]The definition of fusion $n$-category for $n = 2$ is given in [55] in detail, and it will also be reviewed in Section 2.1. For $n \geq 3$, the definition is proposed in [57, Definition II.9], upon the technical assumption mentioned in Remark I.2 of the reference.

sions [33][4]: they constructed explicit two-dimensional classical statistical models acted upon by a given fusion category. The corresponding 1+1d quantum lattice models turn out to be the anyon chain models, which have the given fusion category symmetries. In this paper, we generalize their construction to 2+1 dimensions. Namely, we construct three-dimensional classical statistical models and the corresponding (2+1)-dimensional quantum lattice systems that are acted upon by a given fusion 2-category. We call our 2+1d quantum lattice models the *fusion surface models*, which are (2+1)-dimensional analogues of the 1+1d anyon chains. By construction, the fusion surface models have finite non-invertible symmetries described by general fusion 2-categories, i.e., fusion 2-category symmetries.

The fusion 2-category symmetry in 2+1 dimensions has a particular significance: it includes the symmetry of anyons in topological orders as a special example. Within a generalized Landau paradigm, the existence of anyons in topologically ordered phases can be regarded as a consequence of a spontaneously broken higher (potentially non-invertible) symmetry [69]. Therefore, the fusion surface models with non-invertible higher symmetry provide candidates that might realize a given topological order. In other words, if the model has a gapped point, it is guaranteed that the IR phase contains the anyons we used as an input to the model. While our models include the Levin-Wen string-net models [125] that realize non-chiral topological orders, it probably requires a numerical study to see whether our model can realize chiral topological orders.

Another example of a fusion 2-category symmetry is a finite 2-group (a.k.a. invertible) symmetry with and without an 't Hooft anomaly [2, 3, 5]. A 2-group is a symmetry structure where a conventional symmetry is non-trivially intertwined with an invertible higher symmetry. Our method naturally works for constructing lattice models that possess such a symmetry structure.

In the rest of the introduction, we briefly review the fusion category symmetry in 1+1 dimensions and the Freed-Teleman-Aasen-Fendley-Mong (FT-AFM) construction [33,124].[5] This would serve as a stepping stone to the (2+1)-dimensional case, which is a straightforward generalization of the (1+1)-dimensional case but is apparently more complicated. After reviewing the FT-AFM construction in 1+1d, we will outline the construction of the 2+1d fusion surface models.

## 1.2 Review of the Aasen-Fendley-Mong model

**Fusion category.** In 1+1 dimensions, a finite generalized symmetry is described by a fusion category, which is a generalization of a finite group. It describes the algebraic structure of topological defects of codimension one, or equivalently topological lines, in 1+1 dimensions.[6] More explicitly, a fusion category $\mathcal{C}$ contains the following data (see, e.g., [6, 7] for a detailed explanation and more examples for physicists):

- $\mathrm{Simp}\,\mathcal{C}$: the *finite* set of (isomorphism classes of) "simple objects". A simple object $a \in \mathrm{Simp}\,\mathcal{C}$ represents an oriented indecomposable topological defect line in 1+1d.

---

[4]A statistical-mechanical model, which was defined following a similar conceptual framework, has been previously presented in [124].

[5]We slightly generalize the presentation in [33] in that we allow the multiplicity of fusion coefficients. While such symmetries are not so common in 1+1 dimensions, in 2+1 dimensions there are multiple inequivalent junctions as long as there is a bulk topological line. Thus, we consider the multiplicity in 1+1 dimensions as a warm-up.

[6]When finite one-form symmetries are also present, the whole symmetry is described by a multifusion category, and it is related to the concept of decomposition or "universes" in a (1+1)-dimensional system. See, for example, [32,126,127] for discussions on this point.

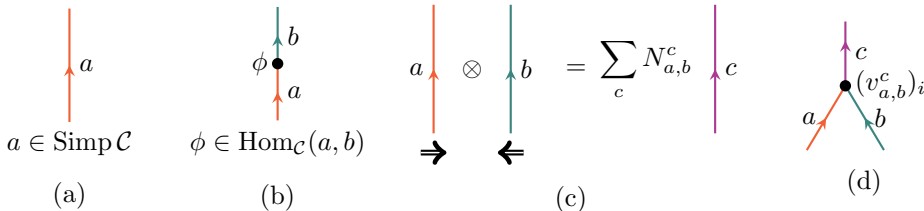

$a \in \mathrm{Simp}\,\mathcal{C}$      $\phi \in \mathrm{Hom}_{\mathcal{C}}(a,b)$

(a)        (b)        (c)        (d)

Figure 2: Data defining a fusion (1-)category $\mathcal{C}$. (a) Topological lines in 1+1 dimensions define objects of $\mathcal{C}$. In particular, simple (i.e., indecomposable) topological lines correspond to simple objects, whereas superpositions of them correspond to non-simple objects. (b) Topological line-changing operators define morphisms between topological lines. (c) The stacking of two topological lines $a$ and $b$ is denoted by $a \otimes b$ and it is in general a superposition of simple topological lines. (d) By locally fusing two lines $a$ and $b$, we get topological junction operators $(v_{a,b}^c)_i$ connecting three lines $a, b$ and $c$. The number of independent junction operators is equal to the fusion coefficient $N_{ab}^c$.

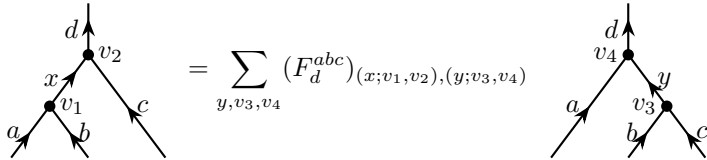

Figure 3: The $F$-move in a fusion category.

There is a special object $I \in \mathrm{Simp}\,\mathcal{C}$ representing the trivial defect. See Figure 2a.

- $\mathrm{Obj}\,\mathcal{C}$: the set of objects. Any object $b \in \mathrm{Obj}\,\mathcal{C}$ takes the form of $b = \bigoplus_{a \in \mathrm{Simp}\,\mathcal{C}} n_a\, a$ where $a$ runs over simple objects and $n_a$'s are non-negative integers. An object $a_1 \oplus a_2$ represents a superposition of defects $a_1$ and $a_2$: the correlation function containing $a_1 \oplus a_2$ is the sum of the correlation function containing $a_1$ and the one containing $a_2$.

- $\mathrm{Hom}_{\mathcal{C}}(a,b)$: the "hom space" between two objects $a$ and $b$. A morphism $\phi \in \mathrm{Hom}_{\mathcal{C}}(a,b)$ from $a$ to $b$ represents a topological line-changing operator connecting the lines $a$ and $b$. See Figure 2b. Such operators form a (finite-dimensional) $\mathbb{C}$-vector space because they can be added and multiplied by complex numbers. In addition, in $\mathrm{Hom}_{\mathcal{C}}(a,a) =: \mathrm{End}_{\mathcal{C}}(a)$, there is the identity operator/morphism $\mathrm{id}_a$. Two line-changing operators $\phi_1 \in \mathrm{Hom}_{\mathcal{C}}(a,b)$ and $\phi_2 \in \mathrm{Hom}_{\mathcal{C}}(b,c)$ can be composed, and the composition defines an element $\phi_2 \circ \phi_1 \in \mathrm{Hom}_{\mathcal{C}}(a,c)$. The hom space between two simple objects $a, b \in \mathrm{Simp}\,\mathcal{C}$ is one dimensional when $a = b$ and zero-dimensional otherwise. In particular, for a simple object $a \in \mathrm{Simp}\,\mathcal{C}$, there is a canonical isomorphism $\mathrm{End}_{\mathcal{C}}(a) \cong \mathbb{C}$, which maps $\lambda\, \mathrm{id}_a \in \mathrm{End}_{\mathcal{C}}(a)$ to $\lambda \in \mathbb{C}$.

- $a \otimes b \in \mathrm{Obj}\,\mathcal{C}$: the tensor product of objects $a$ and $b$. This corresponds to the fusion of topological lines $a$ and $b$. See Figures 2c and 2d. We can expand the tensor product as $a \otimes b = \bigoplus_{c \in \mathrm{Simp}\,\mathcal{C}} N_{ab}^c\, c$, where the non-negative integers $N_{ab}^c$ are called fusion coefficients. As a physicists' convention, we fix a particular (non-canonical) basis $\mathrm{Basis}(a \otimes b, c) := \{(v_{a,b}^c)_i\}_{i=1,\cdots,N_{ab}^c}$ of the hom space $\mathrm{Hom}(a \otimes b, c)$ for $a, b, c \in \mathrm{Simp}\,\mathcal{C}$.

- $F$-symbols $(F_d^{abc})_{(x;v_1,v_2),(y;v_3,v_4)} \in \mathbb{C}$: complex numbers that govern the "$F$-move" depicted in Figure 3. Specifically, the $F$-symbols encode the relationship between

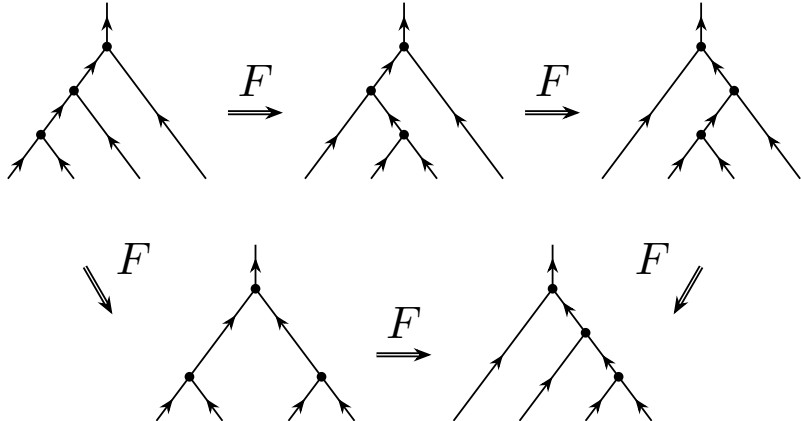

Figure 4: The pentagon identity.

two different ways of composing basis morphisms via the following equation:[7]

$$v_2 \circ (v_1 \otimes \mathrm{id}_c) = \sum_{\substack{y \in \mathrm{Simp}\,\mathcal{C}, \\ v_3 \in \mathrm{Basis}(b \otimes c, y), \\ v_4 \in \mathrm{Basis}(a \otimes y, d)}} (F_d^{abc})_{(x;v_1,v_2),(y;v_3,v_4)} v_4 \circ (\mathrm{id}_a \otimes v_3) \in \mathrm{Hom}(a \otimes b \otimes c, d).$$

(1.1)

The $F$-symbols should satisfy the pentagon identity depicted in Figure 4 [129].

In addition, a fusion category has the following data regarding "dual", which is a relaxed notion of the inverse:

- $a^* \in \mathrm{Obj}\,\mathcal{C}$: the dual of an object $a$. This represents the orientation reversal of $a$, see Figure 5a.

- $\mathrm{ev}_a \in \mathrm{Hom}_{\mathcal{C}}(a^* \otimes a, I)$: the evaluation morphism. This represents the pair-annihilation of topological lines $a$ and $a^*$, see Figure 5b.

- $\mathrm{coev}_a \in \mathrm{Hom}_{\mathcal{C}}(I, a^* \otimes a)$: the coevaluation morphism. This represents the pair-creation of topological lines $a$ and $a^*$, see Figure 5c.[8]

- $\dim(a) \in \mathbb{C}$: the quantum dimension of an object $a$. This quantity is defined by the equality $\mathrm{ev}_a \circ \mathrm{coev}_a = \dim(a)\,\mathrm{id}_I$ and thus corresponds to the vacuum expectation value of a loop of $a$, see Figure 5d. If the quantum dimensions of $a$ and $a^*$ agree with each other for every object $a \in \mathrm{Obj}\,\mathcal{C}$, the fusion category $\mathcal{C}$ is said to be spherical.

The above data that satisfy appropriate consistency conditions define a fusion category [6, 7, 9].

**Examples.** Let us see a few basic examples of fusion categories that naturally appear in physical systems.

- **Finite group.** Topological defects for a finite group symmetry $G$ form the fusion category $\mathrm{Vec}_G$ of $G$-graded vector spaces. The category $\mathrm{Vec}_G$ consists of simple

---

[7]Precisely, the left- and right-hand sides of eq. (1.1) differ by an isomorphism $\alpha_{a,b,c} : (a \otimes b) \otimes c \to a \otimes (b \otimes c)$ called an associator, which is assumed to be the identity here. This assumption is always possible due to Mac Lane strictness theorem [128].

[8]There are left and right evaluation/coevaluation morphisms depending on whether we consider $a \otimes a^*$ or $a^* \otimes a$. For a unitary fusion category, any two of them are automatically determined by the other two because of the unitary structure. See, e.g., [6, 9] for more details.

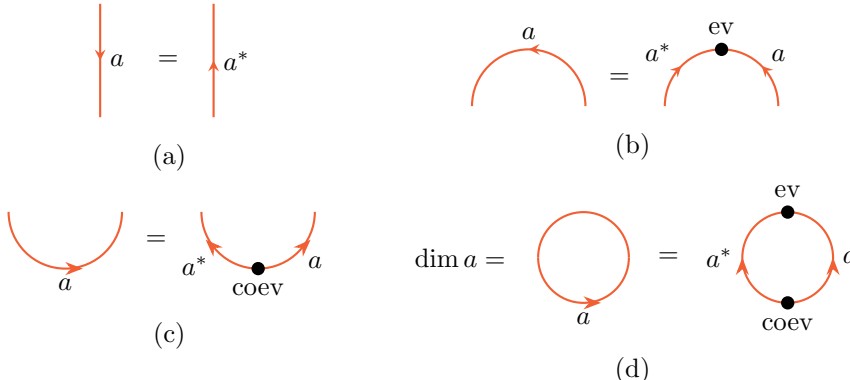

Figure 5: The duality structure in a fusion (1-)category $\mathcal{C}$. (a) The dual $a^*$ of a topological line $a$ is its orientation reversal. (b) There is a morphism $\mathrm{ev}_a : a^* \otimes a \to I$ called evaluation defined by the figure. (c) There is a morphism $\mathrm{coev}_a : I \to a^* \otimes a$ called coevaluation defined by the figure. (d) The expectation value of a loop of a topological line $a$ is called the "quantum dimension" and is denoted by $\dim a$. In the mathematical terms it is the composition $\mathrm{ev}_a \circ \mathrm{coev}_a \in \mathrm{Hom}(I, I) \cong \mathbb{C}$.

objects $L_g$ labeled by group elements $g \in G$. These simple objects obey the group-like fusion rules $L_{g_1} \otimes L_{g_2} = L_{g_1 g_2}$ and have trivial $F$-symbols.[9] The dual of an object $L_g$ is its inverse, i.e., we have $L_g^* = L_{g^{-1}}$. In particular, when $G = \{\mathrm{id}\}$ is the trivial group, $\mathrm{Vec}_G$ reduces to the category $\mathrm{Vec}$ of finite-dimensional vector spaces, which corresponds to the trivial (i.e., no) symmetry. We note that all simple objects of $\mathrm{Vec}_G$ are invertible.

- **Ising category.** A basic example of a non-invertible fusion category arises in the critical Ising model [7, 23, 130]. The category contains simple objects $\eta$ for the $\mathbb{Z}_2$ spin-flip symmetry and $\mathcal{N}$ for the Kramers-Wannier self-duality, forming the fusion category Ising called the "Ising category". The latter object does not constitute a conventional symmetry but does a non-invertible symmetry. The fusion rules of the simple objects are given by

$$\eta \otimes \eta \cong I, \quad \eta \otimes \mathcal{N} \cong \mathcal{N} \otimes \eta \cong \mathcal{N}, \quad \mathcal{N} \otimes \mathcal{N} \cong I \oplus \eta. \tag{1.2}$$

As we can see from the above equation, the simple object $\mathcal{N}$ is indeed non-invertible.

- **Representation category.** Another significant example of a fusion category is the representation category $\mathrm{Rep}(G)$ for a finite group $G$. In this category, simple objects are irreducible representations of $G$, general objects are general finite dimensional representations, morphisms are intertwiners, the tensor product of objects is the ordinary tensor product of representations, and the dual is the complex conjugation. When $G$ is non-abelian, $\mathrm{Rep}(G)$ contains irreducible representations of dimension greater than 1, which are non-invertible.

**Symmetry TFT construction.** In [33], Aasen, Fendley, and Mong constructed both two-dimensional classical statistical mechanical models and (1+1)-dimensional quantum chain models based on fusion categories.

As noted in their paper [33], these models can be naturally understood in terms of three-dimensional topological field theory known as the Turaev-Viro-Barrett-Westbury

---

[9]The $F$-symbols become non-trivial when the finite group symmetry $G$ is anomalous.

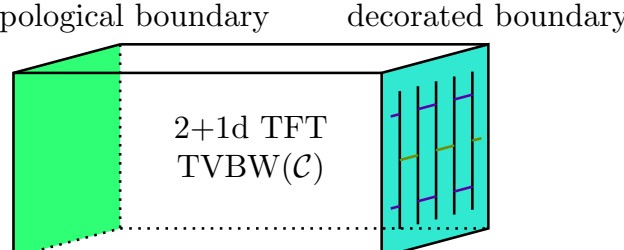

topological boundary      decorated boundary

Figure 6: The AFM height model realized as the TVBW model on a slab with one topological boundary and one non-topological boundary. The non-topological boundary is obtained by decorating the Dirichlet boundary with a defect network as shown in Figure 7.

(TVBW) model [131, 132]. Here, the TVBW model plays the role of what is called "symmetry topological field theory (SymTFT)" in the QFT literature [19, 91, 124, 133–135] and "categorical symmetry" in the condensed matter literature [58, 136–141].[10]

The TVBW model is a state sum model on a (2+1)-dimensional (oriented) spacetime lattice. The input datum of the state sum is a (spherical) fusion category $\mathcal{C}$, and the TVBW model constructed from a fusion category $\mathcal{C}$ is denoted by TVBW($\mathcal{C}$). The model describes the topological order whose anyon data are described by the Drinfeld center $Z(\mathcal{C})$ of $\mathcal{C}$, which is a modular tensor category made out of a fusion category $\mathcal{C}$.

The two-dimensional statistical mechanical model in [33], which we call the AFM height model, can be constructed by placing the TVBW model on a slab, that is, the direct product of an interval $I$ and a two-dimensional oriented closed surface $\Sigma$, see Figure 6. On the left and right boundaries of the slab $I \times \Sigma$, we impose topological and non-topological boundary conditions respectively. The non-topological boundary condition is defined by decorating the "Dirichlet" boundary of TVBW($\mathcal{C}$) with a network of defects as depicted in Figure 7. Here, the Dirichlet boundary condition is a topological boundary condition such that the category of topological lines on the boundary is the input fusion category $\mathcal{C}$. For simplicity, just as in [33, 124], we choose the Dirichlet boundary condition as the topological boundary condition on the left boundary.[11]

To see the symmetry of the AFM height model, we consider the topological lines (or anyons) in TVBW($\mathcal{C}$). Although we can insert any anyons labeled by objects of $Z(\mathcal{C})$ in the 3d bulk, some of them can be absorbed by the topological boundary on the left. Thus, the nontrivial topological lines in the AFM height model are identified with the lines on the topological boundary, which form the fusion category $\mathcal{C}$.[12]

**AFM height model.**  Let us unpack the above abstract construction to obtain explicit 2d classical statistical models. The input data of the AFM height model are listed as follows:

- a (spherical) fusion category $\mathcal{C}$,

---

[10]The idea of using the Turaev-Viro model to construct and study 2d statistical-mechanical systems had already appeared in [124].

[11]In general, topological boundary conditions of TVBW($\mathcal{C}$) are in one-to-one correspondence with (finite semisimple) module categories over $\mathcal{C}$ (equipped with a module trace) [122, 123, 142, 143]. In particular, the Dirichlet boundary condition corresponds to the regular $\mathcal{C}$-module category $\mathcal{C}$.

[12]If the topological boundary condition on the left boundary is the one labeled by a $\mathcal{C}$-module category $\mathcal{M}$, topological lines on the boundary form a fusion category $\mathrm{Fun}_{\mathcal{C}}(\mathcal{M}, \mathcal{M})$, which is the category of $\mathcal{C}$-module endofunctors of $\mathcal{M}$ [122, 123, 142]. Choosing a different topological boundary condition corresponds to gauging (a part of) the fusion category symmetry $\mathcal{C}$ [44, 48, 91, 133].

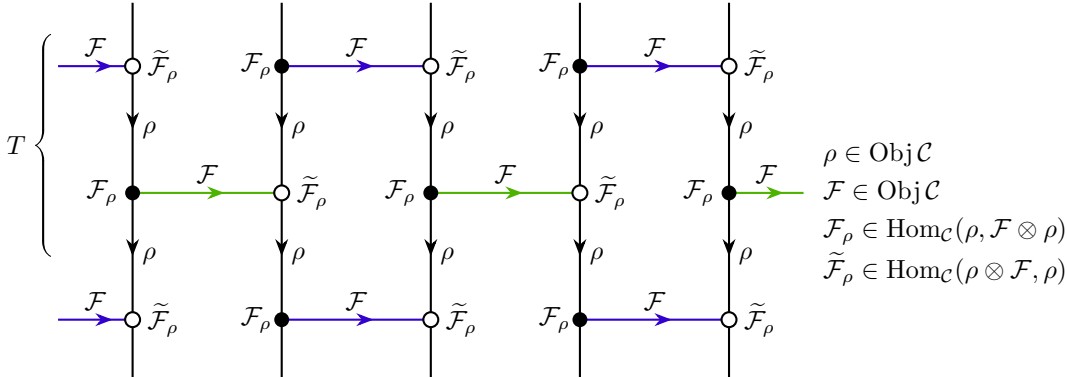

Figure 7: The decorated boundary in Figure 6. The transfer matrix $T$ of the AFM height model is defined by the region indicated in the figure.

- an object $\rho \in \mathrm{Obj}\,\mathcal{C}$,

- an object $\mathcal{F} \in \mathrm{Obj}\,\mathcal{C}$,

- morphisms $\mathcal{F}_\rho \in \mathrm{Hom}_\mathcal{C}(\rho, \mathcal{F} \otimes \rho)$ and $\widetilde{\mathcal{F}}_\rho \in \mathrm{Hom}_\mathcal{C}(\rho \otimes \mathcal{F}, \rho)$.

Note that both $\rho$ and $\mathcal{F}$ are not necessarily simple. Based on the above data,[13] we explicitly describe the AFM height model on a two-dimensional torus $\Sigma = T^2$.

In order to define the model based on the above data, we first draw a defect network on the torus $T^2$ as shown in Figure 7. This defect network plays the role of the spacetime lattice on which dynamical variables reside. Specifically, we assign a dynamical variable $\Gamma_i \in \mathrm{Simp}\,\mathcal{C}$ to each plaquette $i$, and also assign a dynamical variable $\Gamma_{ij} \in \mathrm{Basis}(\Gamma_i \otimes \rho^*, \Gamma_j)$ to each vertical segment (i.e., a black edge) separating plaquettes $i$ and $j$, see Figure 8 for the assignment of these dynamical variables. The dynamical variables $\Gamma_{ij}$ on edges are trivial when all the fusion coefficients $N_{ab}^c$ are either 0 or 1, which is often assumed in the literature for simplicity. The statistical mechanical partition function is defined by the sum of the Boltzmann weights for all possible configurations of dynamical variables:

$$Z = \sum_{\{\Gamma_i\},\{\Gamma_{ij}\}} \prod_h W_h(\vec{\Gamma}). \tag{1.3}$$

Here, $h$ runs over the horizontal segments (i.e., the blue and green edges) in Figure 7, and $W_h(\vec{\Gamma}) \in \mathbb{C}$ is the local Boltzmann weight depending on the dynamical variables around $h$. More specifically, when $h$ is the green edge in Figure 8, the local Boltzmann weight $W_h(\vec{\Gamma})$ depends only on four objects $\{\Gamma_i, \Gamma_j, \Gamma_k, \Gamma_\ell\}$ and four basis morphisms $\{\Gamma_{ij}, \Gamma_{jk}, \Gamma_{i\ell}, \Gamma_{\ell k}\}$ appearing in the same figure. The explicit form of the Boltzmann weight $W_h(\vec{\Gamma})$ is given

---

[13]These data are redundant. In particular, for an invertible element $\phi \in \mathrm{Hom}_\mathcal{C}(\mathcal{F}, \mathcal{F})$, modifying $(\mathcal{F}_\rho, \widetilde{\mathcal{F}}_\rho)$ into $((\phi \otimes \mathrm{id}_\rho) \circ \mathcal{F}_\rho, \widetilde{\mathcal{F}}_\rho \circ (\mathrm{id}_\rho \otimes \phi^{-1}))$ does not change the model. In addition, it turns out that the choice $\mathcal{F} = \rho^* \otimes \rho$ can reproduce the most general model (for a fixed $\rho$). In this case, we can fix the above ambiguity by the condition of, for example, $\widetilde{\mathcal{F}}_\rho = \mathrm{ev}_{\rho^*} \otimes \mathrm{id}_\rho$. With this gauge fixing, the pair $(\rho, \mathcal{F}_\rho)$ parametrizes the model without obvious redundancies except for the overall scaling.

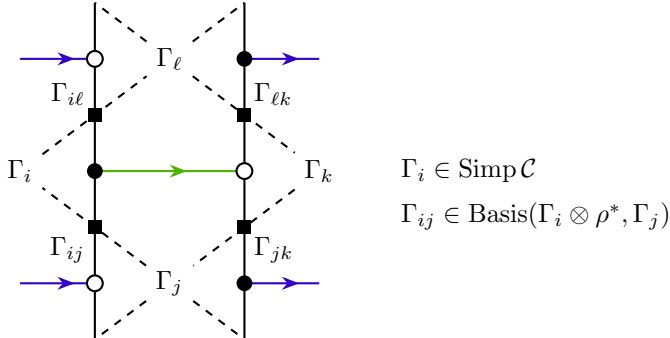

Figure 8: The dynamical variables in the AFM height model.

by the following diagrammatic equation:

$$W_h(\vec{\Gamma}) = \sqrt{\dim \Gamma_j \dim \Gamma_\ell} \times \quad \text{(diagram)} \quad , \tag{1.4}$$

where $\overline{(v_{ab}^c)_i} \in \mathrm{Hom}_{\mathcal{C}}(c, a \otimes b)$ is the dual junction of $(v_{ab}^c)_i \in \mathrm{Basis}(a \otimes b, c)$ that satisfies

$$\mathrm{ev}_c \circ (\mathrm{id}_{c^*} \otimes (v_{ab}^c)_i) \circ (\mathrm{id}_{c^*} \otimes \overline{(v_{ab}^c)_j}) \circ \mathrm{coev}_c = \delta_{ij}. \tag{1.5}$$

We write the basis of $\mathrm{Hom}(c, a \otimes b)$ as $\mathrm{Basis}(c, a \otimes b) := \{\overline{(v_{ab}^c)_i}\}_{i=1,\cdots,N_{ab}^c}$. The weight (1.4) can also be written explicitly in terms of $F$-symbols. If we define the transfer matrix $T$ by the Boltzmann weight on the region indicated in Figure 7, we can write the partition function of the AFM height model as $Z = \mathrm{Tr}\, T^N$, where $N$ is the number of plaquettes in the vertical direction.

**Quantum anyon chain.**  We can obtain a (1+1)-dimensional quantum chain model known as the anyon chain [53, 54] by taking the anisotropic limit of the above two-dimensional statistical mechanical model [33]. The Hilbert space $\mathcal{H}$ of the model is spanned by the fusion trees depicted in Figure 9. Here, we assign a simple object $\Gamma_i \in \mathcal{C}$ to each segment $i$ connecting the vertical $\rho$ lines, and assign a morphism $\Gamma_{i,i+1} \in \mathrm{Basis}(\Gamma_i \otimes \rho^*, \Gamma_{i+1})$ to each vertex connecting the segments $i$, $i+1$ and the vertical $\rho$ line. These simple objects and basis morphisms are the dynamical variables of the model.[14] An assignment of simple objects $\Gamma_i$ and $\Gamma_{i+1}$ is prohibited if the fusion coefficient $N_{\Gamma_i \rho^*}^{\Gamma_{i+1}}$ is zero.

The Hamiltonian $H$ of the model is derived by expanding the transfer matrix of the AFM height model as $T = \mathrm{id}_{\mathcal{H}} - \epsilon H + \mathcal{O}(\epsilon^2)$ in the anisotropic limit, where $\epsilon$ is a small parameter.[15] The Hamiltonian obtained in this way is of the form $H = -\sum_i \hat{h}_{i-1,i,i+1}$,

---

[14]The dimension of the Hilbert space asymptotically grows as $\dim \mathcal{H} \sim (\dim \rho)^L$ where $L \gg 1$ is the number of vertical $\rho$ lines. Thus, we can regard $\rho$ as the degree of freedom at each site. Note that $\dim \rho$ is not necessarily an integer.

[15]As we can see from eq. (1.4), the parameter of the AFM height model is a morphism $\varphi := (\widetilde{\mathcal{F}}_\rho \otimes \mathrm{id}_\rho) \circ (\mathrm{id}_\rho \otimes \mathcal{F}_\rho) \in \mathrm{Hom}(\rho \otimes \rho, \rho \otimes \rho)$. The anisotropic limit is defined by choosing $\varphi = \mathrm{id}_{\rho \otimes \rho} + \epsilon \varphi'$ and taking the limit of $\epsilon \ll 1$. By abuse of notation, $\varphi'$ is also written as $(\widetilde{\mathcal{F}}_\rho \otimes \mathrm{id}_\rho) \circ (\mathrm{id}_\rho \otimes \mathcal{F}_\rho)$ in eq. (1.6).

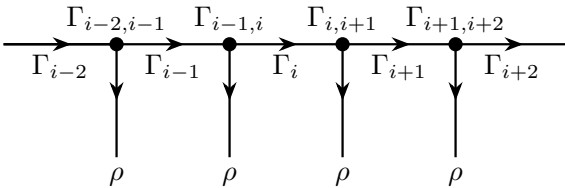

Figure 9: The fusion tree defining the Hilbert space of the quantum anyon chain.

where the local interaction $\hat{h}_{i-1,i,i+1}$ can be expressed diagrammatically as[16]

$$
\hat{h}_{i-1,i,i+1} \quad \vcenter{\hbox{[diagram]}} \quad = \quad \vcenter{\hbox{[diagram]}} \quad = \quad \sum_{\mathcal{F}_{\mathrm{int}}} A(\mathcal{F}_{\mathrm{int}}) \quad \vcenter{\hbox{[diagram]}} .
$$
(1.6)

Here, $\mathcal{F}_{\mathrm{int}} = \{f, v, w\}$ denotes the set consisting of a simple object $f \in \mathrm{Simp}\,\mathcal{C}$ and morphisms $v \in \mathrm{Basis}(\rho, f \otimes \rho)$ and $w \in \mathrm{Basis}(\rho \otimes f, \rho)$ that appear in the diagram on the right-hand side. The weight $A(\mathcal{F}_{\mathrm{int}})$ is a complex number determined by $(\mathcal{F}, \mathcal{F}_\rho, \widetilde{\mathcal{F}}_\rho)$.

The above 1+1d model has a fusion category symmetry $\mathcal{C}$. The symmetry acts on the system "from above" as shown in Figure 10. That is, we define the action of a topological line $a \in \mathrm{Obj}\,\mathcal{C}$ by placing it above the fusion tree and fusing it into the tree using the $F$-move. This symmetry action commutes with the Hamiltonian (1.6) because it acts on the fusion tree "from below":

$$
\left[ \vcenter{\hbox{[diagram]}} \;,\; \vcenter{\hbox{[diagram]}} \right] = 0.
$$
(1.7)

If we write both the Hamiltonian and the symmetry action in terms of the $F$-symbols, the commutation relation (1.7) follows from the pentagon identity shown in Figure 4.

**Examples.** Let us consider several examples of the anyon chain models.

- **Spin chains.** When $\mathcal{C} = \mathrm{Vec}_{\mathbb{Z}_N}$ and $\rho = \bigoplus_{g \in \mathbb{Z}_N} L_g$, the state space of the anyon chain model becomes the tensor product of $N$-dimensional on-site Hilbert spaces. Namely, we have a $\mathbb{Z}_N$-valued spin on each site. The Hamiltonian of the model preserves the on-site $\mathbb{Z}_N$ symmetry that rotates these spins. Thus, the anyon chain model in this case reduces to an ordinary $\mathbb{Z}_N$-symmetric spin chain. More generally, if we choose $\mathcal{C} = \mathrm{Vec}_G$ and $\rho = \bigoplus_{g \in G} L_g$, we obtain a $G$-symmetric spin chain whose on-site Hilbert space is the regular representation of $G$. Furthermore, we can also consider spin chains with anomalous finite group symmetries by choosing $\mathcal{C} = \mathrm{Vec}_G^\omega$, the category of $G$-graded vector spaces with a twist $\omega \in H^3(G, \mathrm{U}(1))$.

---

[16]In the original paper by Aasen, Fendley, and Mong [33], they use a different basis for the local Hamiltonian $\hat{h}_{i-1,i,i+1}$. However, the choice of a basis does not affect the family of Hamiltonians obtained in this way, up to a reparametrization, because the two bases are related by $F$-moves.

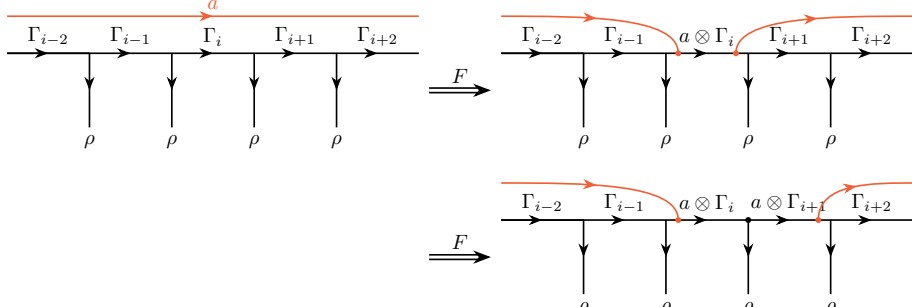

Figure 10: The action of a line $a \in \mathrm{Obj}\,\mathcal{C}$ on the Hilbert space $\mathcal{H}$. This action changes the dynamical variable on each edge from $\Gamma_i$ to $a \otimes \Gamma_i$. The dynamical variables on the vertices are also affected accordingly. The resulting fusion tree can be written as a linear combination of fusion trees whose edges are labeled by fusion channels of $a \otimes \Gamma_i$.

- **Gauged spin chains.** If we choose $\mathcal{C} = \mathrm{Rep}(G)$, we obtain the $G$-gauged version of the spin chains, where the choice of $\rho \in \mathrm{Rep}(G)$ determines the on-site Hilbert space of the ungauged $G$-symmetric spin chain. More specifically, the $\mathrm{Rep}(G)$ symmetry can be ungauged by replacing the Dirichlet boundary condition on the left boundary of the SymTFT with another topological boundary condition labeled by a $\mathrm{Rep}(G)$-module category Vec [44,48,133]. This ungauging procedure results in a $G$-symmetric spin chain whose on-site Hilbert space is $\rho \in \mathrm{Rep}(G)$.

- **Critical Ising model.** When $\mathcal{C}$ is the Ising category and $\rho$ is the Kramers-Wannier duality line, the anyon chain model reproduces the critical Ising model [33].

- **Golden chain.** When $\mathcal{C}$ is the Fibonacci category and $\rho$ is the unique non-invertible line,[17] we obtain the golden chain [53].

- **Haagerup model.** The Haagerup category $\mathcal{H}_3$ is a fusion category that is directly related to neither finite groups nor affine Lie algebras [144–146]. It is generated by a $\mathbb{Z}_3$ invertible line $\eta$ and a self-dual non-invertible line $\rho$ that satisfy the following fusion rules:

$$\rho \otimes \eta \cong \eta^2 \otimes \rho, \quad \rho \otimes \rho \cong I \oplus (I \oplus \eta \oplus \eta^2) \otimes \rho. \tag{1.8}$$

Numerical studies in [40,41] suggest that the anyon chain model and the corresponding statistical mechanical model with the Haagerup symmetry $\mathcal{H}_3$ contain a critical point with central charge $c \sim 2$, but the conclusive identification of the phase is elusive so far.

## 1.3 Generalization to 2+1 dimensions

Our strategy for constructing $(2+1)$-dimensional models, which we call the fusion surface models, is to directly generalize the story reviewed above.

**Fusion 2-category.** In higher dimensions, finite generalized symmetries are expected to be described by higher categories [55–58]. In particular, we can naturally expect that finite generalized symmetries of $(2+1)$-dimensional unitary bosonic systems are generally described by *fusion 2-categories*. The precise definition of a (spherical) fusion 2-category

---

[17]The Fibonacci category consists of two simple objects $I$ and $\tau$ that satisfy the fusion rule $\tau \otimes \tau = I \oplus \tau$.

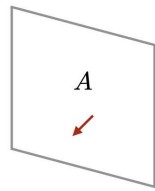 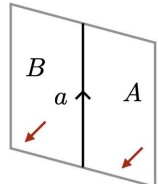 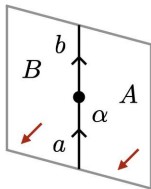

Figure 11: The diagrammatic representations of objects, 1-morphisms, and 2-morphisms in a fusion 2-category. An object $A$ is represented by an oriented surface, whose coorientation is specified by the red arrow perpendicular to the surface. A 1-morphism $a : A \to B$ is represented by an oriented line at the interface between surfaces $A$ and $B$. A 2-morphism $\alpha : a \Rightarrow b$ is represented by a junction of lines $a$ and $b$. In our convention for surface diagrams, a 1-morphism is read from right to left, and a 2-morphism is read from bottom to top. The red arrow specifying the coorientation of a surface will be omitted when the coorientation is clear from the context.

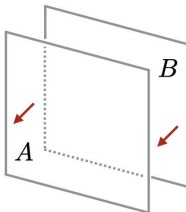

Figure 12: The diagrammatic representation of the tensor product $A \square B$. A surface labeled by $A$ appears in front of a surface labeled by $B$.

can be found in [55]. Here we review the concept of fusion 2-category very briefly. A longer review of fusion 2-categories will be provided in Section 2.1.

In 2+1 dimensions, a defect can have 2-, 1-, or 0-dimensional volume. Correspondingly, a fusion 2-category $\mathcal{C}$ consists of

- $\operatorname{Obj}\mathcal{C}$: the set of objects,
- $\operatorname{Hom}_{\mathcal{C}}(A, B)$: the 1-category of 1-morphisms between objects $A, B \in \operatorname{Obj}\mathcal{C}$, and
- $\operatorname{Hom}_{A \to B}(a, b)$: the vector space of 2-morphisms between 1-morphisms $a, b \in \operatorname{Hom}_{\mathcal{C}}(A, B)$.[18]

As depicted in Figure 11, each element of $\operatorname{Obj}\mathcal{C}$ corresponds to a two-dimensional topological surface, each object of $\operatorname{Hom}_{\mathcal{C}}(A, B)$ corresponds to a topological interface between two surfaces $A$ and $B$, and each element of $\operatorname{Hom}_{A \to B}(a, b)$ corresponds to a topological interface between topological interfaces $a$ and $b$. Note that both 1- and 2-morphisms can be composed, e.g., for $a \in \operatorname{Hom}_{\mathcal{C}}(A, B)$ and $b \in \operatorname{Hom}_{\mathcal{C}}(B, C)$, there exists $b \circ a \in \operatorname{Hom}_{\mathcal{C}}(A, C)$. Similarly, two objects $A$ and $B$ can be stacked on top of each other, which defines the tensor product $A \square B \in \operatorname{Obj}\mathcal{C}$, see Figure 12. Furthermore, a fusion 2-category is also equipped with the duality data such as the dual $A^{\#}$ of an object $A$, the dual $a^*$ of a 1-morphism $a$, and the evaluation and coevaluation morphisms associated with them.

**Symmetry TFT construction.** In [55], a state sum model on a four-dimensional (oriented) spacetime lattice is defined based on a (spherical) fusion 2-category $\mathcal{C}$. We call this state sum model the Douglas-Reutter (DR) model and denote it as $\mathrm{DR}(\mathcal{C})$. The DR model

---

[18]The vector space $\operatorname{Hom}_{A \to B}(a, b)$ is a hom space of the 1-category $\operatorname{Hom}_{\mathcal{C}}(A, B)$.

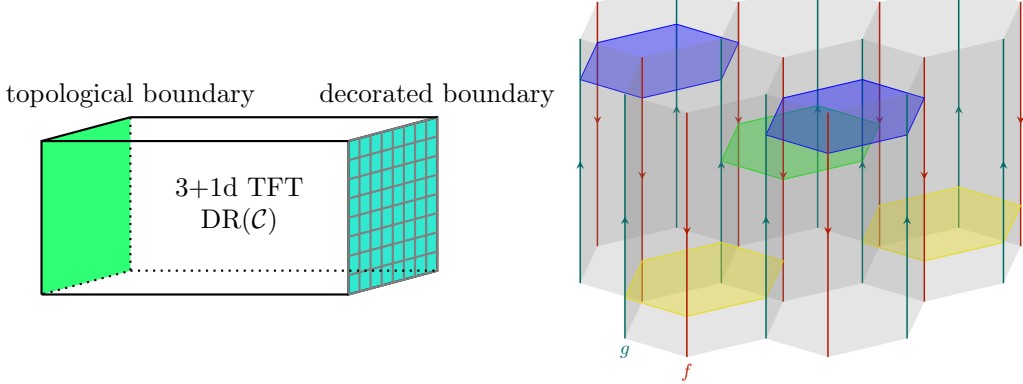

Figure 13: Schematic description of the fusion surface model.

is a four-dimensional version of the TVBW model, and thus we can utilize it to generalize the FT-AFM construction to one higher dimension.

In order to generalize the FT-AFM construction, we consider the DR model $\mathrm{DR}(\mathcal{C})$ on a four-dimensional slab $I \times T^3$ where $I$ is an interval and $T^3$ is a three-dimensional torus, see the left panel of Figure 13. On the left boundary of the slab, we impose the Dirichlet boundary condition of $\mathrm{DR}(\mathcal{C})$.[19] In particular, this means that topological defects on the left boundary are described by the fusion 2-category $\mathcal{C}$ that we started with. On the other hand, on the right boundary, we impose a non-topological boundary condition that is obtained by decorating the Dirichlet boundary with the defect network shown in the right panel of Figure 13.

Since the bulk of $\mathrm{DR}(\mathcal{C})$ is topological, the configuration depicted in the left panel of Figure 13 defines a purely 3d classical statistical model, which we call the 3d height model. Furthermore, by taking the anisotropic limit of the 3d height model, we can define the corresponding 2+1d quantum lattice model, which we call the fusion surface model. The derivation of the 3d height models and 2+1d fusion surface models will be explained in detail in sections 3 and 4 respectively. In the rest of this subsection, we will briefly summarize the definition of the 2+1d fusion surface model and describe its fusion 2-category symmetry.

### 1.3.1 Fusion surface models

**Input data.** The fusion surface model is a 2+1d quantum model on a honeycomb lattice.[20] In order to define the state space of this model, we fix the following data, see Figure 14:

- a (spherical) fusion 2-category $\mathcal{C}$,
- objects $\rho, \sigma, \lambda \in \mathrm{Obj}\,\mathcal{C}$,
- 1-morphisms $f \in \mathrm{Hom}_{\mathcal{C}}(\rho \square \sigma, \lambda)$ and $g \in \mathrm{Hom}_{\mathcal{C}}(\sigma \square \rho, \lambda)$.

Moreover, to define the Hamiltonian, we fix the data listed below, see also Figure 15:

---

[19]More generally, we can also use a different topological boundary condition on the left boundary, which should be labeled by a module 2-category over $\mathcal{C}$. A different choice of a module 2-category would correspond to a different way of gauging the fusion 2-category symmetry $\mathcal{C}$. In the case of a non-anomalous finite group symmetry $2\mathrm{Vec}_G$, the relation between the choice of a module 2-category and the (twisted) gauging is studied in [113]. We do not explore this generalization in this paper.

[20]Although a honeycomb lattice is convenient for our purpose, e.g. because at each vertex the minimal number (three) of edges meet, it should be straightforward to generalize the model to another lattice.

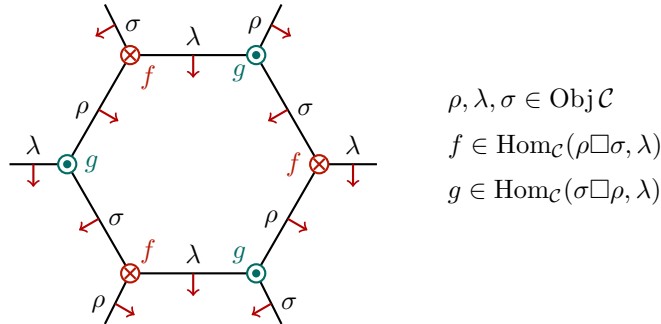

Figure 14: Data defining the state space.

$$\rho, \lambda, \sigma \in \mathrm{Obj}\,\mathcal{C}$$
$$f \in \mathrm{Hom}_{\mathcal{C}}(\rho \square \sigma, \lambda)$$
$$g \in \mathrm{Hom}_{\mathcal{C}}(\sigma \square \rho, \lambda)$$

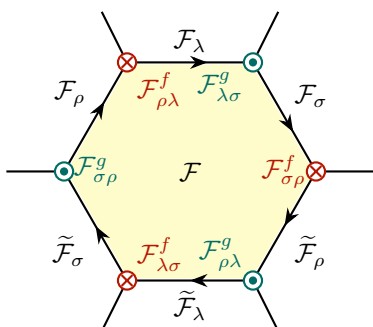

Figure 15: Data defining the Hamiltonian. The domains/codomains of the symbols are summarized in eqs. (1.9) and (1.10).

- an object $\mathcal{F} \in \mathrm{Obj}\,\mathcal{C}$,

- 1-morphisms

$$
\begin{aligned}
\mathcal{F}_\lambda \in \mathrm{Hom}_{\mathcal{C}}(\mathcal{F}\square\lambda, \lambda), \quad &\mathcal{F}_\rho \in \mathrm{Hom}_{\mathcal{C}}(\mathcal{F}\square\rho, \rho), \quad \mathcal{F}_\sigma \in \mathrm{Hom}_{\mathcal{C}}(\mathcal{F}\square\sigma, \sigma), \\
\widetilde{\mathcal{F}}_\lambda \in \mathrm{Hom}_{\mathcal{C}}(\lambda\square\mathcal{F}, \lambda), \quad &\widetilde{\mathcal{F}}_\rho \in \mathrm{Hom}_{\mathcal{C}}(\rho\square\mathcal{F}, \rho), \quad \widetilde{\mathcal{F}}_\sigma \in \mathrm{Hom}_{\mathcal{C}}(\sigma\square\mathcal{F}, \sigma),
\end{aligned}
\tag{1.9}
$$

- 2-morphisms

$$
\begin{aligned}
\mathcal{F}_{\lambda\sigma}^f &\in \mathrm{Hom}_{\rho\square\sigma\square\mathcal{F}\to\lambda}(\widetilde{\mathcal{F}}_\lambda \circ (f\square\mathbf{1}_\mathcal{F}), f\circ(\mathbf{1}_\rho\square\widetilde{\mathcal{F}}_\sigma)), \\
\mathcal{F}_{\sigma\rho}^g &\in \mathrm{Hom}_{\sigma\square\mathcal{F}\square\rho\to\lambda}(g\circ(\widetilde{\mathcal{F}}_\sigma\square\mathbf{1}_\rho), g\circ(\mathbf{1}_\sigma\square\mathcal{F}_\rho)), \\
\mathcal{F}_{\rho\lambda}^f &\in \mathrm{Hom}_{\mathcal{F}\square\rho\square\sigma\to\lambda}(f\circ(\mathcal{F}_\rho\square\mathbf{1}_\sigma), \mathcal{F}_\lambda\circ(\mathbf{1}_\mathcal{F}\square f)), \\
\mathcal{F}_{\lambda\sigma}^g &\in \mathrm{Hom}_{\mathcal{F}\square\sigma\square\rho\to\lambda}(\mathcal{F}_\lambda\circ(\mathbf{1}_\mathcal{F}\square g), g\circ(\mathcal{F}_\sigma\square\mathbf{1}_\rho)), \\
\mathcal{F}_{\sigma\rho}^f &\in \mathrm{Hom}_{\rho\square\mathcal{F}\square\sigma\to\lambda}(f\circ(\mathbf{1}_\rho\square\mathcal{F}_\sigma), f\circ(\widetilde{\mathcal{F}}_\rho\square\mathbf{1}_\sigma)), \\
\mathcal{F}_{\rho\lambda}^g &\in \mathrm{Hom}_{\sigma\square\rho\square\mathcal{F}\to\lambda}(g\circ(\mathbf{1}_\sigma\square\widetilde{\mathcal{F}}_\rho), \widetilde{\mathcal{F}}_\lambda\circ(g\square\mathbf{1}_\mathcal{F})).
\end{aligned}
\tag{1.10}
$$

**State space.** The state space $\mathcal{H}_0$ of the fusion surface model on a honeycomb lattice is a specific subspace of a larger state space $\mathcal{H}$ that is spanned by fusion diagrams of the

following form:

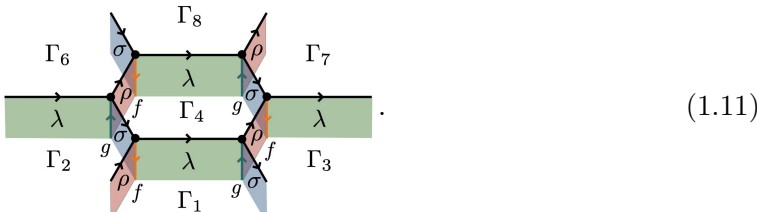

$$\text{(1.11)}$$

Dynamical variables living on plaquettes (written in white), edges (written in black), and vertices (written in black) of the honeycomb lattice are labeled by simple objects, simple 1-morphisms, and basis 2-morphisms of $\mathcal{C}$ respectively. More precisely, the plaquette variables take values in the set of representatives of connected components of simple objects of $\mathcal{C}$, and the edge variables take values in the set of representatives of isomorphism classes of simple 1-morphisms of $\mathcal{C}$, see Section 2.1 for the terminology. The dynamical variables on plaquettes are denoted by $\Gamma_i$ in the above equation, whereas the dynamical variables on edges and vertices are not specified in order to avoid cluttering the diagram. The colored surfaces in eq. (1.11) are labeled by objects $\rho$, $\sigma$, and $\lambda$, which are not dynamical variables of the model. Similarly, the colored edges in eq. (1.11) are labeled by 1-morphisms $f$ and $g$, which are not dynamical variables as well. The state space $\mathcal{H}_0$ is the subspace of $\mathcal{H}$ on which the eigenvalue of the plaquette operator $\hat{B}_p$ defined by the following equation is 1 for every plaquette $p$:

$$\hat{B}_p \quad \cdots \quad = \sum_{\Gamma_{45} \in \text{End}(\Gamma_4)} \frac{\dim(\Gamma_{45})}{\text{Dim}(\Gamma_4)} \quad \cdots \quad . \tag{1.12}$$

Here, $\dim(\Gamma_{45})$ is the quantum dimension of a simple 1-morphism $\Gamma_{45}$, and $\text{Dim}(\Gamma_4) := \dim(\Gamma_4) \dim(\text{End}_{\mathcal{C}}(\Gamma_4))$ is the product of the quantum dimension of a simple object $\Gamma_4$ and the total dimension of a fusion 1-category $\text{End}_{\mathcal{C}}(\Gamma_4) := \text{Hom}_{\mathcal{C}}(\Gamma_4, \Gamma_4)$, see Section 2 for the definitions of these quantities. The diagram on the right-hand side of eq. (1.12) is evaluated by fusing the loop labeled by $\Gamma_{45}$ to the edges of the honeycomb lattice. We note that $\hat{B}_p$ is a local commuting projector just like the plaquette operator of the Levin-Wen model [125]. The projector to the subspace $\mathcal{H}_0$ is given by the product of $\hat{B}_p$'s on all plaquettes, namely, we have

$$\mathcal{H}_0 = \left( \prod_{p:\text{ plaquettes}} \hat{B}_p \right) \mathcal{H}. \tag{1.13}$$

**Hamiltonian.** The Hamiltonian of the model is given by $H = -\sum_{p:\text{ plaquettes}} \hat{T}_p$, where each term $\hat{T}_p$ is defined by the following diagrammatic equation:

$$\hat{T}_p \quad \cdots \quad = \quad \cdots \quad . \tag{1.14}$$

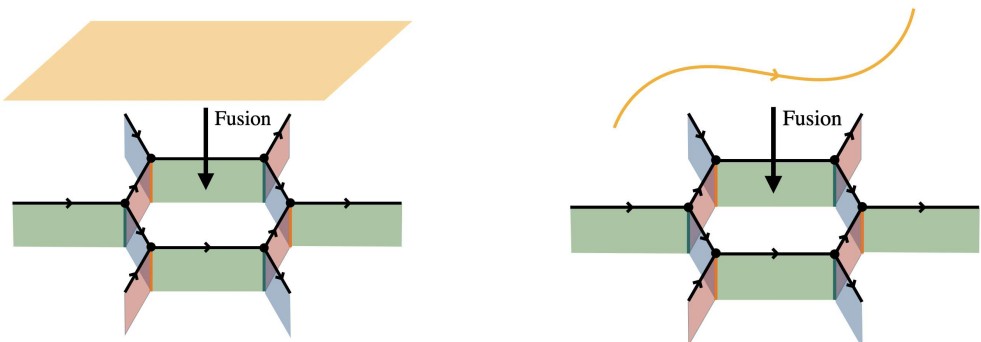

Figure 16: The action of a fusion 2-category symmetry is defined by the fusion of surface and line defects to the spatial lattice.

The diagram on the right-hand side is evaluated by fusing the yellow surface, yellow edges, and yellow vertices into the honeycomb lattice. Here, they are labeled by an object $\mathcal{F} \in \mathrm{Obj}\,\mathcal{C}$, 1-morphisms (1.9), and 2-morphisms (1.10) as shown in Figure 15. If we expand them in terms of simple objects, simple 1-morphisms, and basis 2-morphisms, the Hamiltonian (1.14) can also be written as

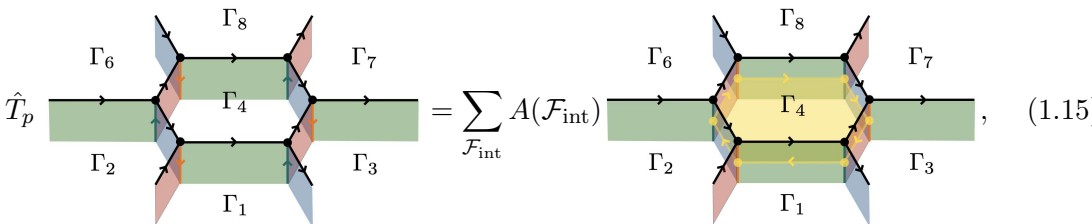

$$\hat{T}_p \quad = \quad \sum_{\mathcal{F}_{\mathrm{int}}} A(\mathcal{F}_{\mathrm{int}}) \qquad , \qquad (1.15)$$

where the weight $A(\mathcal{F}_{\mathrm{int}})$ is a complex number, and the summation on the right-hand side is taken over all possible simple objects, simple 1-morphisms, and basis 2-morphisms labeling the yellow surface, yellow edges, and yellow vertices in the diagram. The labels summed over are collectively denoted by $\mathcal{F}_{\mathrm{int}}$ in the above equation. Specifically, $\mathcal{F}_{\mathrm{int}}$ consists of one simple object, six simple 1-morphisms, and six basis 2-morphisms. Under several assumptions that we spell out in Section 4.2, we can show that the Hamiltonian (1.15) becomes Hermitian if the weight satisfies $\mathcal{A}(\mathcal{F}_{\mathrm{int}}) = A(\overline{\mathcal{F}}_{\mathrm{int}})^*$, where $\overline{\mathcal{F}}_{\mathrm{int}}$ basically means the dual of $\mathcal{F}_{\mathrm{int}}$, see Section 4.2 for more details.[21]

**Symmetry.** The fusion surface model defined above has a fusion 2-category symmetry described by the input fusion 2-category $\mathcal{C}$. The action of the symmetry is defined by the operation of fusing topological surfaces and topological lines into the honeycomb lattice from above as shown in Figure 16. This symmetry action can be written in terms of the 10-j symbols of the fusion 2-category. The commutativity of the symmetry action and the Hamiltonian (1.15) is guaranteed by the coherence conditions on the 10-j symbols. This is because the symmetry operator acts on a state from above, while the Hamiltonian acts on a state from below, see eq. (1.7) for an analogous statement in 1+1d.

**Examples.** Let us see several examples of the fusion surface model that we will discuss in this paper.

---

[21]We can always make the Hamiltonian Hermitian by adding the Hermitian conjugate, which would not violate the fusion 2-category symmetry of the model.

- **Spin models with anomalous finite group symmetries.** When the input fusion 2-category is the 2-category $2\mathrm{Vec}_G^\omega$ of $G$-graded 2-vector spaces with a twist $\omega \in H^4(G, \mathrm{U}(1))$ [55], the fusion surface model has a finite group symmetry $G$ with an anomaly $\omega$. In particular, when $\rho, \sigma$, and $\lambda$ are the sum of all simple objects $\rho = \sigma = \lambda = \boxplus_{g \in G} g$, the dynamical variables of the model are $G$-valued spins on all plaquettes. Thus, the fusion surface model in this case reduces to an ordinary spin model with an anomalous finite group symmetry. We will study this example in Section 4.4.3. As a special case, the fusion surface model includes the anomaly free $G$-symmetric spin model discussed in [113].

- **Lattice models with non-invertible and invertible 1-form symmetries.** When the input fusion 2-category is (the condensation completion of) a ribbon category $\mathcal{B}$,[22] the fusion surface model has a non-invertible 1-form symmetry described by $\mathcal{B}$. We will discuss this example briefly in Section 4.4.2. In particular, when the fusion rules of $\mathcal{B}$ are group-like, the fusion surface model reduces to an ordinary spin model with an anomalous invertible 1-form symmetry. This example will be discussed in more detail in Section 5.1.

- **Kitaev honeycomb model without a magnetic field.** When the input fusion 2-category is (the condensation completion of) the Ising category, we can obtain the Kitaev honeycomb model without a magnetic field [147] as a variant of the fusion surface model. We will consider this example in Section 5.2.

- **Non-chiral topological phases with fusion 2-category symmetries.** For any fusion 2-category $\mathcal{C}$, we can construct a commuting projector Hamiltonian with $\mathcal{C}$ symmetry by defining the input data of the fusion surface model using a separable algebra in $\mathcal{C}$. Since the Hamiltonian is the sum of local commuting projectors, this model would realize a non-chiral topological phase with $\mathcal{C}$ symmetry. We expect that all non-chiral topological phases with arbitrary fusion 2-category symmetries can be realized in this way by choosing a separable algebra appropriately. This example will be discussed in Section 5.3.

## 1.4 Structure of the paper

This paper is organized as follows. In Section 2, we review fusion 2-categories and the state sum construction of the 4d Douglas-Reutter TFT. In Section 3, we define the 3d height models on a cubic lattice, which are three-dimensional analogues of the 2d AFM height models. In Section 4, we derive the 2+1d fusion surface models on a honeycomb lattice by taking an appropriate limit of the 3d height models. In particular, we see that the fusion surface models are (2+1)-dimensional analogues of the 1+1d anyon chain models. We also investigate the unitarity and fusion 2-category symmetries of these models. Finally, in Section 5, we study several examples of the fusion surface models, including those that would realize general non-chiral topological phases with fusion 2-category symmetries.

# 2 Preliminaries

Throughout the paper, we suppose that the base field of a fusion 2-category is $\mathbb{C}$.

---

[22]The condensation completion physically means that we add in topological surfaces consisting of the condensates of topological lines. Mathematically, the condensation completion of a ribbon category $\mathcal{B}$ is described by the 2-category $\mathrm{Mod}(\mathcal{B})$ of module categories over $\mathcal{B}$.

## 2.1 Fusion 2-categories

Finite symmetries in 2+1 dimensions are characterized by the algebraic structure of topological surfaces, topological lines, and topological point defects. In unitary theories, these defects are expected to form a spherical fusion 2-category. In this section, we briefly review the basics of fusion 2-categories. We refer the reader to [55] for more details, see also [148].

A fusion 2-category $\mathcal{C}$ consists of objects, 1-morphisms between objects, and 2-morphisms between 1-morphisms. The 1-morphisms between objects $A$ and $B$ form a finite semisimple 1-category, which is denoted by $\text{Hom}_{\mathcal{C}}(A, B)$. The subscript $\mathcal{C}$ of $\text{Hom}_{\mathcal{C}}(A, B)$ will often be omitted in what follows. The 2-morphisms between 1-morphisms $a$ and $b$ form a finite dimensional vector space, which is denoted by $\text{Hom}(a, b)$. Physically, objects, 1-morphisms, and 2-morphisms correspond to surface defects, line defects, and point defects respectively. The diagrammatic representations of these data are shown in Figure 11.

The fusion of topological surfaces labeled by $A$ and $B$ corresponds to taking the tensor product of objects $A$ and $B$. The tensor product of $A$ and $B$, which is denoted by $A \square B$, is represented by the diagram of layered two surfaces, where the surface labeled by $A$ is put in front of the surface labeled by $B$, see Figure 12. The unit of the tensor product is called a unit object and is denoted by $I$. The unit object $I$ corresponds to a trivial surface defect, which is represented by an invisible diagram.

The fusion of topological lines labeled by $a$ and $b$ corresponds to the composite of 1-morphisms $a$ and $b$, which is denoted by $a \circ b$. In the diagrammatic representation of the composite $a \circ b$, the line labeled by $a$ is on the left of the line labeled by $b$. There is also a similar correspondence between the fusion of topological point defects and the composition of 2-morphisms.

Every object and every 1-morphism of a fusion 2-category $\mathcal{C}$ have their duals. The dual $A^{\#}$ of an object $A$ is represented by the orientation reversal of a surface diagram labeled by $A$. Similarly, the dual $a^*$ of a 1-morphism $a$ is represented by the orientation reversal of a line labeled by $a$.[23] We note that $a^*$ is a 1-morphism from $B$ to $A$ when $a$ is a 1-morphism from $A$ to $B$. Taking the duals of objects and 1-morphisms is involutive, i.e., we have $A^{\#\#} = A$ and $a^{**} = a$.

An object $A \in \mathcal{C}$ is called a simple object if the vector space of 2-endomorphisms of the identity 1-morphism $\mathbf{1}_A$ is one-dimensional, i.e., $\text{End}(\mathbf{1}_A) := \text{Hom}(\mathbf{1}_A, \mathbf{1}_A) \cong \mathbb{C}$. The unit object $I$ of a fusion 2-category is required to be simple. Similarly, a 1-morphism $a \in \text{Hom}(A, B)$ is called a simple 1-morphism if its endomorphism space $\text{End}(a)$ is a one-dimensional vector space. We note that the identity 1-morphism of a simple object is simple. A fusion 2-category $\mathcal{C}$ has only finitely many (isomorphism classes of) simple objects and simple 1-morphisms between simple objects.

Any objects and 1-morphisms in $\mathcal{C}$ can be decomposed into finite direct sums of simple objects and simple 1-morphisms respectively. The direct sum of objects $A$ and $B$ is denoted by $A \boxplus B$, whereas the direct sum of 1-morphisms $a$ and $b$ is denoted by $a \oplus b$. Simple objects and simple 1-morphisms are indecomposable, which means that they cannot be decomposed into direct sums any further.

Since the unit object $I$ is simple, the identity 1-morphism $\mathbf{1}_I$ has a one-dimensional vector space of 2-endomorphisms $\text{End}(\mathbf{1}_I) \cong \mathbb{C}$. This implies that any 2-endomorphism of $\mathbf{1}_I$ is proportional to the identity 2-morphism and hence can be identified with a number. In particular, a closed surface diagram, when viewed as a 2-endomorphism of $\mathbf{1}_I$, gives rise to a complex number. This enables us to define complex numbers $\dim_R(a)$ and $\dim_L(a)$

---

[23]In the mathematical literature, the dual of a 1-morphism is often called the adjoint.

for a 1-morphism $a \in \mathrm{Hom}(A, B)$ by the following sphere diagrams:

$$\dim_R(a) = \vcenter{\hbox{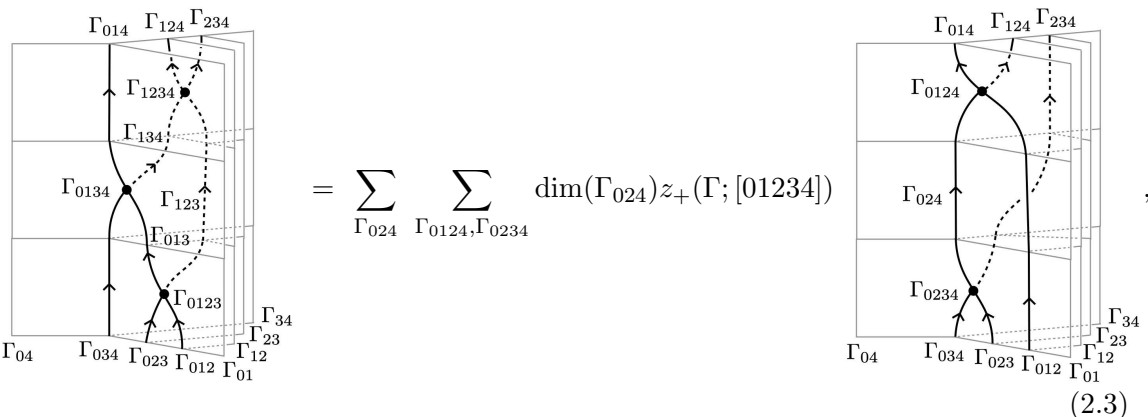}} , \qquad \dim_L(a) = \quad . \tag{2.1}$$

We call these quantities the right and left dimensions of $a$. Here, we implicitly used the (co)evaluation 1- and 2-morphisms to define the cups and caps as in the definition of the quantum dimension in a fusion 1-category, cf. Figure 5d.

The left and right dimensions agree with each other when a fusion 2-category $\mathcal{C}$ is equipped with an additional structure called a pivotal structure. In a pivotal fusion 2-category, the right (or equivalently left) dimension of a 1-morphism $a$ is simply denoted by $\dim(a)$ and is called the quantum dimension of $a$. In particular, the quantum dimension of the identity 1-morphism $\mathbf{1}_A \in \mathrm{End}(A)$ is denoted by $\dim(A) := \dim(\mathbf{1}_A)$ and is called the quantum dimension of $A$.[24] A pivotal fusion 2-category $\mathcal{C}$ is said to be spherical if the quantum dimension of every object agrees with the quantum dimension of its dual. In the rest of this paper, a fusion 2-category $\mathcal{C}$ always means a spherical fusion 2-category.

The quantum dimension of a 1-morphism $a \in \mathrm{Hom}(A, B)$ can be understood as the trace of the identity 2-morphism of $a$. More generally, the trace of a 2-morphism $\alpha \in \mathrm{End}(a)$ is defined as the value of the following sphere diagram:

$$\mathrm{Tr}(\alpha) = \quad . \tag{2.2}$$

The trace defines a non-degenerate pairing $\langle \alpha, \beta \rangle := \mathrm{Tr}(\alpha\beta)$ between 2-morphisms $\alpha \in \mathrm{Hom}(a, b)$ and $\beta \in \mathrm{Hom}(b, a)$. The dual bases of the vector spaces $\mathrm{Hom}(a, b)$ and $\mathrm{Hom}(b, a)$ with respect to the above non-degenerate pairing are denoted by $\{\alpha_i\}$ and $\{\overline{\alpha_i}\}$, which satisfy $\langle \alpha_i \overline{\alpha}_j \rangle = \delta_{ij}$. We call $\alpha_i$ and $\overline{\alpha}_i$ basis 2-morphisms or normalized 2-morphisms.

In a fusion 1-category, the associativity of the tensor product is captured by the $F$-symbols. In a fusion 2-category, the tensor product among objects satisfies a higher associativity, which is captured by the data called the 10-j symbols. Specifically, the 10-j symbols $z_+(\Gamma; [01234])$ and $z_-(\Gamma; [01234])$ are defined by the following diagrammatic equations:[25]

$$\quad = \sum_{\Gamma_{024}} \sum_{\Gamma_{0124}, \Gamma_{0234}} \dim(\Gamma_{024}) z_+(\Gamma; [01234]) \quad , \tag{2.3}$$

<hr>

[24]Simple objects $A$ and $B$ can have different quantum dimensions even if they are isomorphic to each other. Physically, isomorphic objects with different quantum dimensions differ by an invertible 2d TFT.

[25]The 10-j symbols defined here are the same as those defined in [55]. Indeed, eq. (2.3) and (2.4) reduce to the original definition of the 10-j symbols given in [55] if we take the trace of these equations after post-composing a 2-morphism that is the dual of a summand on the right-hand side.

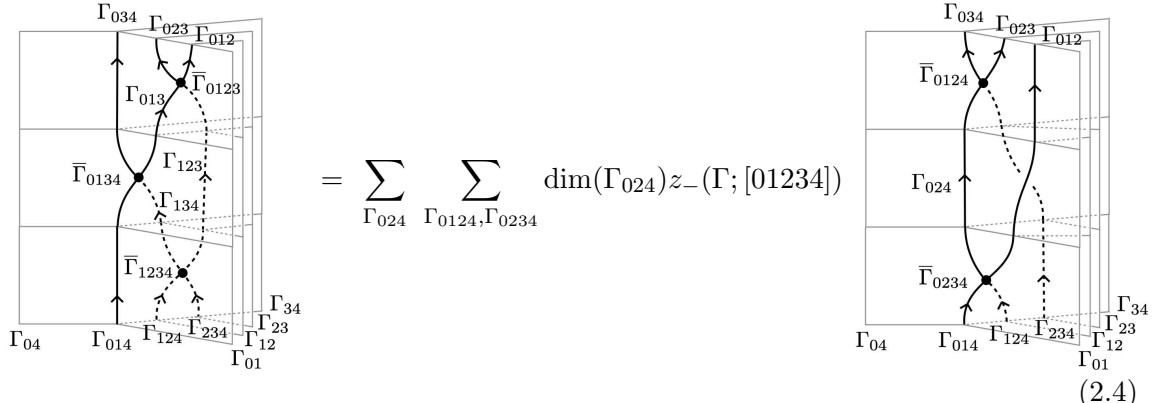

$$(2.4)$$

Here, the diagram in the above equation consists of ten surfaces $[ij]$ labeled by simple objects $\Gamma_{ij}$ where $0 \leq i < j \leq 4$, ten lines $[ijk]$ labeled by simple 1-morphisms $\Gamma_{ijk} \in \mathrm{Hom}_{\mathcal{C}}(\Gamma_{ij}\square\Gamma_{jk}, \Gamma_{ik})$ where $0 \leq i < j < k \leq 4$, and five points $[ijkl]$ labeled by basis 2-morphisms $\Gamma_{ijkl} \in \mathrm{Hom}(\Gamma_{ikl} \circ (\Gamma_{ijk}\square\mathbf{1}_{\Gamma_{kl}}), \Gamma_{ijl} \circ (\mathbf{1}_{\Gamma_{ij}}\square\Gamma_{jkl}))$ where $0 \leq i < j < k < l \leq 4$. The summation on the right-hand side is taken over (isomorphism classes of) simple 1-morphisms $\Gamma_{024}$ and basis 2-morphisms $\Gamma_{0124}$ and $\Gamma_{0234}$. The braiding of two lines $\Gamma_{012}$ and $\Gamma_{234}$ on the right-hand side represents the interchanger 2-isomorphism [55], which reduces to the ordinary braiding isomorphism when $\Gamma_{012}$ and $\Gamma_{234}$ are 1-endomorphisms of the unit object $I$.

For later convenience, we define the notion of connected components of simple objects and simple 1-morphisms in a fusion 2-category $\mathcal{C}$. Simple objects $A$ and $B$ are connected if and only if there exists a non-zero 1-morphism between them, and we say they are in the same connected component. Similarly, simple 1-morphisms $a$ and $b$ are connected if and only if there exists a non-zero 2-morphism between them. We note that the connected component of a simple object is bigger than the isomorphism class of the simple object.[26] This is because there can be non-zero 1-morphisms between simple objects $A$ and $B$ even if they are not isomorphic to each other. The set of connected components of simple objects in $\mathcal{C}$ is denoted by $\pi_0\mathcal{C}$. By a slight abuse of notation, we will also write the set of representatives of connected components as $\pi_0\mathcal{C}$.

## 2.2 Douglas-Reutter TFT

The Douglas-Reutter TFT is a four-dimensional oriented topological field theory obtained from a spherical fusion 2-category $\mathcal{C}$ [55]. This TFT generalizes various 4d TFTs known in the literature, see table 1.[27] In this section, we review the Douglas-Reutter TFT, which is denoted by $\mathrm{DR}(\mathcal{C})$, following Walker's universal state sum [160].

Let $M$ be a closed oriented 4-manifold. In order to define the partition function of the Douglas-Reutter TFT $\mathrm{DR}(\mathcal{C})$ on $M$, we first choose a triangulation of $M$. We also give a branching structure on the triangulated 4-manifold $M$ by choosing a global order $o$ of 0-simplices.

The dynamical variables of the Douglas-Reutter TFT $\mathrm{DR}(\mathcal{C})$ are simple objects $\Gamma_{ij}$ living on 1-simplices $[ij]$, simple 1-morphisms $\Gamma_{ijk}$ living on 2-simplices $[ijk]$, and basis

---

[26]On the other hand, the connected component of a simple 1-morphism agrees with the isomorphism class. This is because a simple 1-morphism $a : A \to B$ in a fusion 2-category is a simple object in a finite semisimple 1-category $\mathrm{Hom}(A, B)$ and every non-zero morphism between simple objects in such a 1-category is an isomorphism due to Schur's lemma.

[27]There are also other 4d TFTs such as unoriented TFTs [149,150], spin and pin TFTs [151], and TFTs with U(1) symmetry [152], which are not examples of the Douglas-Reutter TFT. We do not consider these TFTs in this paper. It would be interesting to generalize our analyses to these TFTs.

Table 1: The relation between the Douglas-Reutter TFT and other 4d TFTs [55].

| Fusion 2-category | Douglas-Reutter TFT |
|---|---|
| Finite group (with a twist) | (twisted) Dijkgraaf-Witten TFT [153] |
| Finite 2-group (with a twist) | (twisted) Yetter TFT [154, 155] |
| Ribbon category | Crane-Yetter TFT [156–158] |
| $G$-crossed braided fusion category | Cui TFT [159] |

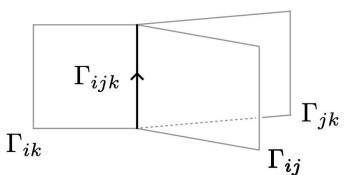 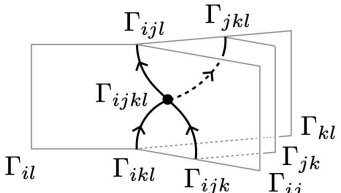

Figure 17: The compatibility conditions on a configuration of dynamical variables. The left figure represents the compatibility between the dynamical variable on a 2-simplex $[ijk]$ and those on the boundary of $[ijk]$. The right figure represents the compatibility between the dynamical variable on a 3-simplex $[ijkl]$ and those on the boundary of $[ijkl]$.

2-morphisms $\Gamma_{ijkl}$ living on 3-simplices $[ijkl]$, where $i < j < k < l$ with respect to the global order $o$ of 0-simplices. Here, $\Gamma_{ij}$ and $\Gamma_{ijk}$ are taken from the set of representatives of connected components of simple objects and simple 1-morphisms. A configuration of the above dynamical variables must be compatible with the monoidal structure of a fusion 2-category $\mathcal{C}$ in the following sense: a simple 1-morphism $\Gamma_{ijk}$ is a 1-morphism from $\Gamma_{ij}\square\Gamma_{jk}$ to $\Gamma_{ik}$ and a basis 2-morphism $\Gamma_{ijkl}$ is a 2-morphism from $\Gamma_{ikl}\circ(\Gamma_{ijk}\square\mathbf{1}_{\Gamma_{kl}})$ to $\Gamma_{ijl}\circ(\mathbf{1}_{\Gamma_{ij}}\square\Gamma_{jkl})$. These compatibility conditions can be expressed by the fusion diagrams shown in Figure 17. A configuration $\Gamma$ of dynamical variables is referred to as a $\mathcal{C}$-state.

The partition function of $\mathrm{DR}(\mathcal{C})$ is given by the sum of appropriate weights over all possible $\mathcal{C}$-states. More specifically, the partition function $Z_{\mathrm{DR}}(M)$ on a closed oriented 4-manifold $M$ is defined by the following formula: [55, 160]

$$Z_{\mathrm{DR}}(M) = \sum_{\Gamma} \prod_{\text{0-simplices } [i]} \frac{1}{\dim(\mathcal{C})} \prod_{\text{1-simplices } [ij]} \frac{1}{\mathrm{Dim}(\Gamma_{ij})}$$
$$\prod_{\text{2-simplices } [ijk]} \dim(\Gamma_{ijk}) \prod_{\text{4-simplices } [ijklm]} z_{\epsilon_o([ijklm])}(\Gamma; [ijklm]). \tag{2.5}$$

Here, $\dim(\mathcal{C}) = \sum_{X\in\pi_0\mathcal{C}}(\dim(\mathrm{End}(X)))^{-1}$ is the total dimension of a fusion 2-category $\mathcal{C}$,[28] $\mathrm{Dim}(\Gamma_{ij}) := \dim(\Gamma_{ij})\dim(\mathrm{End}(\Gamma_{ij}))$ is the product of the quantum dimension $\dim(\Gamma_{ij})$ of a simple object $\Gamma_{ij}$ and the global dimension $\dim(\mathrm{End}(\Gamma_{ij}))$ of the endomorphism 1-category $\mathrm{End}(\Gamma_{ij})$, and $\dim(\Gamma_{ijk})$ is the quantum dimension of a simple 1-morphism $\Gamma_{ijk}$. The weight $z_{\epsilon_o([ijklm])}(\Gamma; [ijklm])$ on a 4-simplex $[ijklm]$ is the 10-j symbol defined by eqs. (2.3) and (2.4). The subscript $\epsilon_o([ijklm])$ is a sign determined by the orientation of a 4-simplex $[ijklm]$ in the following way: $\epsilon_o([ijklm])$ is $+$ if the orientation of $[ijklm]$ induced by the orientation of the underlying manifold $M$ agrees with the one induced by the global order $o$ of 0-simplices, and $\epsilon_o([ijklm])$ is $-$ otherwise.

---

[28]The global dimension $\dim(\mathrm{End}(X))$ of $\mathrm{End}(X)$ for a simple object $X \in \pi_0\mathcal{C}$ is given by the sum of the squared norm $\|x\|^2 = (\dim(x)/\dim(X))^2$ for all (isomorphism classes of) simple objects $x \in \mathrm{End}(X)$. We note that $\|x\|$ is the quantum dimension of $x$ viewed as an object of a fusion 1-category $\mathrm{End}(X)$.

The formula (2.5) is based on Walker's universal state sum construction [160], which is slightly different from the original formulation by Douglas and Reutter [55]. In the original paper by Douglas and Reutter, dynamical variables $\Gamma_{ij}$ on 1-simplices are taken from the set of isomorphism classes of simple objects rather than the set of connected components of them. Accordingly, the scalar factor on each 1-simplex $[ij]$ is further divided by the number of (isomorphism classes of) simple objects in the connected component of $\Gamma_{ij}$. In the subsequent sections, we will use Walker's universal state sum construction instead of the original formulation by Douglas and Reutter because the former can immediately be applied to manifolds with general cell decompositions, which is convenient for our purposes.

In order to apply Walker's universal state sum to a closed 4-manifold $M$ with a general cell decomposition, we begin with turning the cell decomposition into a handle decomposition by thickening each cell. A thickened $j$-cell is called a $j$-handle, which has the shape of $B^j \times B^{4-j}$ where $B^n$ is an $n$-dimensional ball. We sometimes use the terms $j$-cell and $j$-handle interchangeably. The boundary of a $j$-cell is topologically a 3-sphere $S^3$ consisting of two regions $S^{j-1} \times B^{4-j}$ and $B^j \times S^{3-j}$. The former region is glued to $i$-cells for $i < j$, while the latter region is glued to $i$-cells for $i > j$. Given a handle decomposition as above, we can write down the partition function of the Douglas-Reutter TFT on $M$ as

$$Z_{\mathrm{DR}}(M) = \sum_{\Gamma} \prod_{\text{4-handles } h_4} \frac{1}{\dim(\mathcal{C})} \prod_{\text{3-handles } h_3} \frac{1}{\mathrm{Dim}(\Gamma(h_3))} \\ \prod_{\text{2-handles } h_2} \mathrm{ev}(\Gamma(\partial h_2)) \prod_{\text{0-handles } h_0} \mathrm{ev}(\Gamma(\partial h_0)), \tag{2.6}$$

where a $\mathcal{C}$-state $\Gamma$ is a compatible assignment of simple objects $\Gamma(h_3)$, simple 1-morphisms $\Gamma(h_2)$, and basis 2-morphisms $\Gamma(h_1)$ to 3-handles $h_3$, 2-handles $h_2$, and 1-handles $h_1$ respectively. In the above equation, $\Gamma(\partial h_j)$ represents the surface diagram that appears as the intersection of the 3-sphere $\partial h_j$ and the original (i.e., not thickened) cell decomposition labeled by a $\mathcal{C}$-state $\Gamma$. Since the surface diagram $\Gamma(\partial h_j)$ is closed, it defines a 2-morphism from the unit object $I$ to itself, which is canonically identified with a complex number. This complex number, which would be expressed in terms of the 10-j symbols and the quantum dimensions, is denoted by $\mathrm{ev}(\Gamma(\partial h_j))$ in the above equation. We note that eq. (2.6) reduces to eq. (2.5) when the cell decomposition is the dual of a triangulation.

We can also apply Walker's universal state sum to manifolds with boundaries. In order to compute the partition function on an oriented 4-manifold $M$ with boundary $\partial M$, we endow $M$ with a cell decomposition and label the boundary 2-cells, boundary 1-cells, and boundary 0-cells by simple objects, simple 1-morphisms, and basis 2-morphisms in a consistent manner. The labeling on the boundary cells defines a fusion diagram on $\partial M$, which we denote by $\mathcal{F}$. A fusion diagram $\mathcal{F}$ on the boundary is called a coloring. When the boundary $\partial M$ is colored by a fusion diagram $\mathcal{F}$, a $\mathcal{C}$-state $\Gamma$ in the bulk is constrained so that the dynamical variable $\Gamma(h_j)$ on a bulk $j$-cell intersecting the boundary $\partial M$ agrees with the label on the boundary $(j-1)$-cell $h_j \cap \partial M$. For this setup, the partition function on an oriented 4-manifold $M$ with a non-empty boundary can be written as

$$Z_{\mathrm{DR}}(M; \mathcal{F}) = \mathcal{N}_{\mathcal{F}} \sum_{\Gamma} \prod_{\text{4-handles } h_4} \frac{1}{\dim(\mathcal{C})} \prod_{\text{3-handles } h_3} \frac{1}{\mathrm{Dim}(\Gamma(h_3))} \\ \prod_{\text{2-handles } h_2} \mathrm{ev}(\Gamma(\partial h_2)) \prod_{\text{0-handles } h_0} \mathrm{ev}(\Gamma(\partial h_0)), \tag{2.7}$$

where the products on the right-hand side are taken over $j$-handles that do not intersect the boundary. The numerical factor $\mathcal{N}_{\mathcal{F}}$ is a complex number that depends only on the

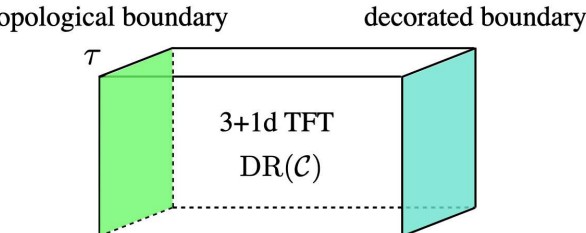

Figure 18: A schematic picture of the 3d classical statistical model obtained from the 4d Douglas-Reutter TFT on a slab $N \times [0,1]$. We impose a topological boundary condition $\tau$ on the left boundary $N \times \{0\}$. On the other hand, we impose a (not necessarily topological) boundary condition decorated by a coloring $\sum_{\mathcal{F}} A(\mathcal{F})\mathcal{F}$ on the right boundary $N \times \{1\}$. The above geometry defines a genuine three-dimensional system because the four-dimensional bulk is topological.

coloring $\mathcal{F}$ on the boundary. In particular, $\mathcal{N}_{\mathcal{F}}$ does not depend on the cell decomposition of the bulk. The detailed definition of $\mathcal{N}_{\mathcal{F}}$ does not matter for later applications as we will see shortly in Section 3.1.

# 3 3d height models

## 3.1 3d classical statistical models from 4d Douglas-Reutter TFT

We construct three-dimensional classical statistical models based on the Douglas-Reutter TFT. Specifically, we put the DR theory on a slab $N \times [0,1]$, where $N$ is a closed oriented 3-manifold equipped with a cell decomposition. See Figure 18. The partition function of this 3d classical statistical model is given by

$$Z(A;\tau) = \sum_{\mathcal{F}} A(\mathcal{F}) Z_{\mathrm{DR}}(N \times [0,1]; \tau, \mathcal{F}), \quad A(\mathcal{F}) \in \mathbb{C}. \tag{3.1}$$

Here, $Z_{\mathrm{DR}}(N \times [0,1]; \tau, \mathcal{F})$ is the partition function of the Douglas-Reutter TFT on a slab $N \times [0,1]$, where $\tau$ and $\mathcal{F}$ specify the boundary conditions on the left and right boundaries respectively. Concretely, $\tau$ is a topological boundary condition on the left boundary $N \times \{0\}$ and $\mathcal{F}$ is a coloring on the right boundary $N \times \{1\}$.[29] We note that the numerical factor $\mathcal{N}_{\mathcal{F}}$ in eq. (2.7) is absorbed into the redefinition of the weight $A(\mathcal{F})$ in eq. (3.1) and therefore the precise form of $\mathcal{N}_{\mathcal{F}}$ does not matter.

The above partition function defines a purely three-dimensional system because we can squash the bulk TFT due to its topological nature. Indeed, we can write the right-hand side of the above equation without using the four-dimensional bulk at all. To see this, we choose a cell decomposition of $N \times [0,1]$ so that every $j$-cell in the bulk is of the form $c_{j-1} \times [0,1]$, where $c_{j-1}$ is a $(j-1)$-cell on the boundary. In particular, there are no 0-cells in the bulk. For this choice of a cell decomposition, we can think of a $\mathcal{C}$-state, which is originally defined as a collection of dynamical variables in the bulk, as a collection of dynamical variables on the boundary $N \times \{1\}$ by projecting the dynamical variables on $j$-cells in the bulk onto the corresponding $(j-1)$-cells on the boundary $N \times \{1\}$. More specifically, a $\mathcal{C}$-state assigns a simple object, a simple 1-morphism, and a basis 2-morphism

---

[29]We expect that a topological boundary $\tau$ is obtained by decorating the Dirichlet boundary by a fine mesh of topological surfaces labeled by an algebra object of the input fusion 2-category. In the subsequent sections, we will restrict our attention to the case where $\tau$ is the Dirichlet boundary.

to each 0-cell, 1-cell, and 2-cell on the boundary $N \times \{1\}$. We note that the dynamical variables are now living only on the boundary $N \times \{1\}$. Since the projection of dynamical variables in the bulk to the boundary preserves the locality, the partition function (3.1) can be viewed as a genuine 3d classical statistical model.

The symmetry of the above 3d classical statistical model is determined by the pair of the 4d TFT $DR(\mathcal{C})$ in the bulk and a topological boundary condition $\tau$ on the left boundary. For this reason, the bulk topological field theory is called a symmetry TFT [19, 91, 124, 133–135] or categorical symmetry [58, 136–141] in the literature. Specifically, the symmetry of the 3d model is generated by topological defects living on the topological boundary $\tau$. Therefore, a different choice of a topological boundary condition gives rise to a different symmetry.[30]

In what follows, we take $\tau$ to be the Dirichlet boundary condition, which means that the coloring on the left boundary $N \times \{0\}$ is a trivial fusion diagram. In this case, eq. (2.7) implies that we can shrink the left boundary $N \times \{0\}$ to a point when computing the partition function because shrinking the Dirichlet boundary to a point does not affect the surface diagrams $\Gamma(\partial h_j)$ in eq. (2.7). More specifically, the partition function on a slab $N \times [0, 1]$ agrees, up to a constant $\mathcal{N}_0$, with the partition function on a cone $\text{pt} * N$, which is a (singular) manifold obtained by shrinking the Dirichlet boundary to a point. Thus, the partition function of the 3d classical statistical model can be written as

$$Z(A) = \sum_{\mathcal{F}} A(\mathcal{F}) Z_{\text{DR}}(\text{pt} * N; \mathcal{F}). \tag{3.2}$$

We note that the scalar factor $\mathcal{N}_0$ is absorbed into the redefinition of $A(\mathcal{F})$.

Before proceeding, we emphasize that the simple objects, simple 1-morphisms, and basis 2-morphisms contained in the coloring $\mathcal{F}$ are regarded as dynamical variables of the 3d model for the time being. These dynamical variables will be integrated out when we will define the 3d height models in the next subsection. Hence, the dynamical variables of the 3d height models consist only of simple objects, simple 1-morphisms, and basis 2-morphisms contained in the $\mathcal{C}$-state $\Gamma$.

## 3.2 3d height models on a triangulated cubic lattice

Let us now explicitly construct the 3d height model based on the above general idea of constructing 3d classical statistical models. The lattice $\Lambda$ on which the 3d height model is defined is a cubic lattice endowed with a triangulation and a branching structure as shown in Figure 19. The underlying 3-manifold of $\Lambda$ is supposed to be a 3-torus $T^3$.

A configuration of dynamical variables is specified by a pair $(\Gamma, \mathcal{F})$ of a $\mathcal{C}$-state $\Gamma$ and a coloring $\mathcal{F}$. As we mentioned in the previous subsection, a $\mathcal{C}$-state $\Gamma$ assigns a simple object $\Gamma_i$ to each 0-simplex $[i]$, a simple 1-morphism $\Gamma_{ij}$ to each 1-simplex $[ij]$, and a basis 2-morphism $\Gamma_{ijk}$ to each 2-simplex $[ijk]$. On the other hand, a coloring $\mathcal{F}$ consists of a simple object $\mathcal{F}_{ij}$ on each 1-simplex $[ij]$, a simple 1-morphism $\mathcal{F}_{ijk}$ on each 2-simplex $[ijk]$, and a basis 2-morphism $\mathcal{F}_{ijkl}$ on each 3-simplex $[ijkl]$.[31] This difference is because, in the 4d Douglas-Reutter theory on the cone, while $\Gamma$'s are assigned to the cells connecting the vertex $[\text{pt}]$ and the right boundary, $\mathcal{F}$'s are assigned to the simplices on the boundary.

---

[30]Given a topological boundary condition, we can obtain another topological boundary condition by condensing a separable algebra formed by a set of topological defects on the boundary. The condensation of a separable algebra on the topological boundary is regarded as the gauging of the fusion 2-category symmetry of the 3d model.

[31]A coloring $\mathcal{F}$ gives rise to a fusion diagram on the dual cell decomposition of $\Lambda$.

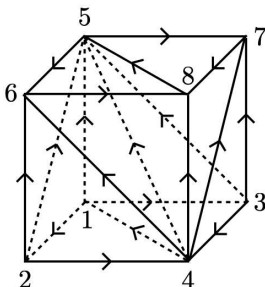

Figure 19: The cubic lattice on which the 3d height model is defined is equipped with the above triangulation and branching structure.

The partition function of a general 3d classical statistical model defined in the previous subsection can be written as the sum of the Boltzmann weights over all possible configurations of dynamical variables on $\Lambda$ as follows:

$$Z(A) = \sum_\Gamma \sum_\mathcal{F} A(\mathcal{F}) \prod_{\text{0-simplices } [i]} \frac{1}{\text{Dim}(\Gamma_i)} \prod_{\text{1-simplices } [ij]} \dim(\Gamma_{ij}) \prod_{\text{cubes } c} z(\Gamma, \mathcal{F}; c). \quad (3.3)$$

Here, the weight $z(\Gamma, \mathcal{F}; c)$ on a cube $c$ is given by the product of the weights on the 3-simplices contained in $c$. By construction, the weight on a 3-simplex $[ijkl]$ is the 10-j symbol on the corresponding 4-simplex pt $* [ijkl]$, whose vertices are ordered as pt $<$ $i, j, k, l$. Therefore, if we label the vertices of a cube $c$ by $1, 2, \cdots, 8$ as shown in Figure 19, we can write the weight $z(\Gamma, \mathcal{F}; c)$ as

$$\begin{aligned} z(\Gamma, \mathcal{F}; c) = {} & z_\epsilon(\Gamma, \mathcal{F}; \text{pt} * [1245]) z_\epsilon(\Gamma, \mathcal{F}; \text{pt} * [2456]) z_\epsilon(\Gamma, \mathcal{F}; \text{pt} * [4568]) \\ & z_{-\epsilon}(\Gamma, \mathcal{F}; \text{pt} * [1345]) z_{-\epsilon}(\Gamma, \mathcal{F}; \text{pt} * [3457]) z_{-\epsilon}(\Gamma, \mathcal{F}; \text{pt} * [4578]), \end{aligned} \quad (3.4)$$

where $\epsilon = \pm$ is a sign determined by the choice of an orientation of the underlying manifold $T^3$. We note that the relative signs for different 3-simplices $[ijkl]$ are determined solely by the branching structure on $\Lambda$, which is independent of the choice of the orientation of $T^3$. For example, the relative sign for 3-simplices $[1245]$ and $[2456]$ can be computed as follows. We first suppose that each 4-simplex pt $* [ijkl]$ has an orientation $\epsilon_{ijkl}$.[32] In this case, a 4-simplex pt $* [1245]$ induces an orientation $-\epsilon_{1245}$ on a 3-simplex pt $* [245]$, whereas a 4-simplex pt $* [2456]$ induces an orientation $\epsilon_{2456}$ on pt $* [245]$. These induced orientations must be opposite to each other because the underlying (singular) manifold pt $* T^3$ is oriented. Therefore, we find $\epsilon_{1245} = \epsilon_{2456}$, which shows that the relative sign for $[1245]$ and $[2456]$ is positive. Similarly, we can compute the relative signs for other 3-simplices. We can also check that $\epsilon$ in eq. (3.4) does not depend on a cube $c$ by computing the relative signs for 3-simplices contained in adjacent cubes.

A 3d height model is obtained by choosing a weight $A(\mathcal{F})$ appropriately as we describe below. In order to define the 3d height model, we first take $A(\mathcal{F})$ to be the product of local weights on cubes:

$$A(\mathcal{F}) = \prod_{\text{cubes } c} A_c(\mathcal{F}^c). \quad (3.5)$$

We note that the function $A_c(\mathcal{F}^c)$ can depend on a cube $c$, meaning that the Boltzmann weight can be non-uniform on the lattice $\Lambda$. The argument $\mathcal{F}^c$ denotes the set of dynamical variables contained in the coloring on a cube $c$. More specifically, $\mathcal{F}^c$ consists of 19 simple

---

[32]We define the orientation of an $n$-simplex $[i_0 \cdots i_{n-1}]$ to be positive if it is an even permutation of $i_0, \cdots, i_{n-1}$. Otherwise, the orientation of $[i_0 \cdots i_{n-1}]$ is defined to be negative.

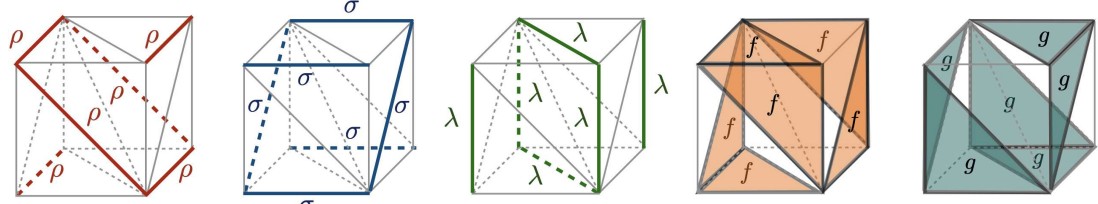

Figure 20: We fix the coloring on the boundary of each cube as above, while we keep the coloring inside each cube dynamical. The dual fusion diagram of the above coloring is shown in Figure 21.

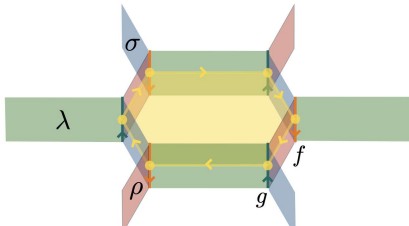

Figure 21: The coloring (3.6) on a triangulated cube defines a fusion diagram on the dual cell decomposition, which looks like a hexagonal prism as shown above. The yellow surface, lines and points in the above diagram are the duals of the internal edge, triangles and tetrahedra of a triangulated cube.

objects $\mathcal{F}_{ij}^c$ on 1-simplices $[ij]$ in $c$, 12 simple 1-morphisms $\mathcal{F}_{ijk}^c$ on 2-simplices $[ijk]$ in $c$, and 6 basis 2-morphisms $\mathcal{F}_{ijkl}^c$ on 3-simplices $[ijkl]$ in $c$. The superscript $c$ will be omitted when it is clear from the context. Furthermore, we fix the coloring on the boundary of each cube so that we can integrate out the coloring $\mathcal{F}$ later while preserving the locality of the Boltzmann weight. In other words, we require that a local weight $A_c(\mathcal{F}^c)$ is non-zero only when a coloring $\mathcal{F}^c$ satisfies the following conditions, see also Figure 20:

$$
\begin{aligned}
\mathcal{F}_{12}^c &= \mathcal{F}_{34}^c = \mathcal{F}_{56}^c = \mathcal{F}_{78}^c = \mathcal{F}_{35}^c = \mathcal{F}_{46}^c = \rho, \\
\mathcal{F}_{13}^c &= \mathcal{F}_{24}^c = \mathcal{F}_{57}^c = \mathcal{F}_{68}^c = \mathcal{F}_{25}^c = \mathcal{F}_{47}^c = \sigma, \\
\mathcal{F}_{15}^c &= \mathcal{F}_{26}^c = \mathcal{F}_{37}^c = \mathcal{F}_{48}^c = \mathcal{F}_{14}^c = \mathcal{F}_{58}^c = \lambda, \\
\mathcal{F}_{124}^c &= \mathcal{F}_{125}^c = \mathcal{F}_{347}^c = \mathcal{F}_{357}^c = \mathcal{F}_{468}^c = \mathcal{F}_{568}^c = f, \\
\mathcal{F}_{134}^c &= \mathcal{F}_{135}^c = \mathcal{F}_{246}^c = \mathcal{F}_{256}^c = \mathcal{F}_{478}^c = \mathcal{F}_{578}^c = g.
\end{aligned}
\tag{3.6}
$$

Here, $\rho$, $\sigma$, and $\lambda$ are simple objects, and $f : \rho\square\sigma \to \lambda$ and $g : \sigma\square\rho \to \lambda$ are simple 1-morphisms, all of which are chosen arbitrarily.[33] We emphasize that the choice of these simple objects and simple 1-morphisms does not depend on a cube $c$. The above coloring defines a fusion diagram on the dual of a triangulated cube as shown in Figure 21. The coloring inside a cube $c$, which is denoted by $\mathcal{F}_{\mathrm{int}}^c$, remains dynamical after fixing the coloring on the boundary of $c$. Since the coloring $\mathcal{F}_{\mathrm{int}}^c$ inside $c$ can be chosen independently of the coloring $\mathcal{F}_{\mathrm{int}}^{c'}$ inside any other cube $c'$, the summation over all colorings $\mathcal{F}$ in eq. (3.3) can be factorized into the summations over $\mathcal{F}_{\mathrm{int}}^c$ for all cubes $c$. Therefore, we can write the partition function (3.3) as

$$
Z(A) = \sum_{\Gamma} \prod_{[i]} \frac{1}{\mathrm{Dim}(\Gamma_i)} \prod_{[ij]} \dim(\Gamma_{ij}) \prod_c \sum_{\mathcal{F}_{\mathrm{int}}^c} A_c(\mathcal{F}_{\mathrm{int}}^c) z(\Gamma, \mathcal{F}; c).
\tag{3.7}
$$

---

[33]Although we assume that objects $\rho, \sigma, \lambda$, and 1-morphisms $f, g$ are simple, a similar derivation of the 3d height model can be applied even when they are non-simple.

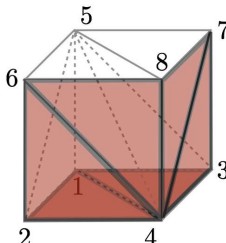 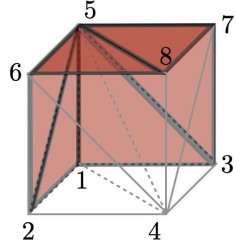

Figure 22: The initial time slice (left) and the final time slice (right) on a single cube.

For later convenience, we write the above partition function in a more compact form as

$$Z(A) = \sum_{\Gamma} \prod_{c:\text{ cubes}} W_c(\Gamma; c), \tag{3.8}$$

where the Boltzmann weight $W_c(\Gamma; c)$ on a cube $c$ is defined by

$$W_c(\Gamma; c) = \dim(\Gamma_{45}^c) \sqrt{\prod_{j=4,5} \frac{\prod_{i=1,2,3} \prod_{k=6,7,8} \dim(\Gamma_{ij}^c) \dim(\Gamma_{jk}^c)}{\text{Dim}(\Gamma_j^c)}} \sum_{\mathcal{F}_{\text{int}}^c} A_c(\mathcal{F}_{\text{int}}^c) z(\Gamma, \mathcal{F}; c). \tag{3.9}$$

We call the 3d classical statistical model defined by the above partition function a 3d height model because this model is a three-dimensional analogue of the 2d AFM height model in [33]. We note that a coloring $\mathcal{F}$ is already integrated out in eq. (3.8) and hence is no longer regarded as a dynamical variable of the 3d height model.

Although we do not describe in detail, we can also incorporate topological defects by inserting them on the left (i.e., topological) boundary of the Douglas-Reutter theory before squashing the four-dimensional bulk. These topological defects generate the symmetry of the 3d height model. In Section 4.4, we will see how this symmetry is realized in the corresponding 2+1d quantum model.

# 4 2+1d fusion surface models

Throughout this section, we suppose that the weight $A_c(\mathcal{F}_{\text{int}}^c)$ does not depend on cube $c$ and write it simply as $A(\mathcal{F}_{\text{int}}^c)$.

## 4.1 2+1d fusion surface models from 3d height models

In this subsection, we derive the Hamiltonian of the 2+1d fusion surface model on a honeycomb lattice, which is the quantum counterpart of the 3d height model on the triangulated cubic lattice $\Lambda$. To this end, we first choose a time direction on $\Lambda$. The time direction on each cube is given by the direction from vertex [4] to vertex [5]. We call vertices [4] and [5] the initial vertex and the final vertex respectively. The above choice of a time direction enables us to define the initial time slice and the final time slice for each cube as follows: the initial time slice consists of the faces containing the initial vertex [4], whereas the final time slice consists of the faces containing the final vertex [5], see Figure 22.[34] A global time slice on the whole cubic lattice is illustrated in Figure 23, where the cubes are colored in blue, green, and yellow for later convenience.

---

[34]We can equally choose the time direction in the opposite way, and we will end up with the same (family of) models.

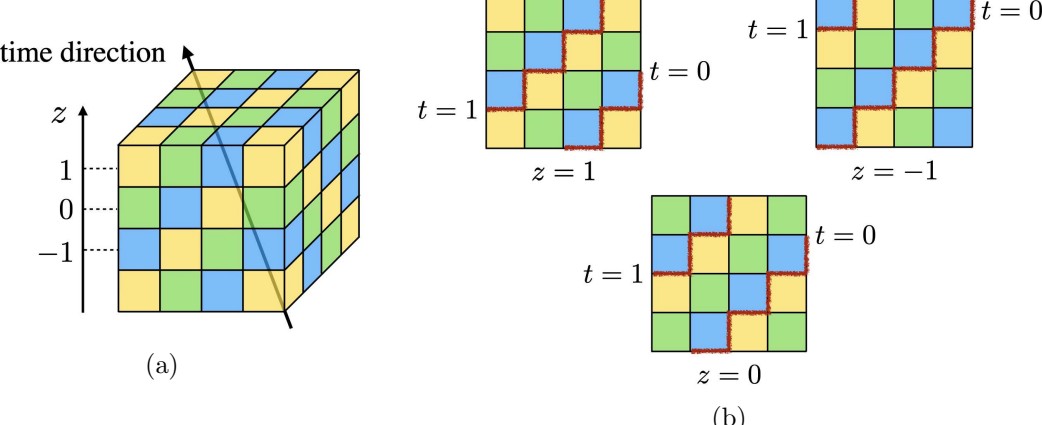

Figure 23: (a) A colored cubic lattice. A blue cube is surrounded by three green cubes and three yellow cubes. A green cube is surrounded by three yellow cubes and three blue cubes. A yellow cube is surrounded by three blue cubes and three green cubes. (b) The initial and final time slices at $t = 0, 1$ viewed from the $z$ direction. Each red line represents the intersection of a time slice and a plane perpendicular to the $z$-axis.

The triangulation of a cubic lattice shown in Figure 19 gives rise to a triangular lattice on a single time slice, which is the Poincaré dual of a honeycomb lattice. We illustrate the relation between a triangulated cubic lattice, a triangular lattice, and a honeycomb lattice in Figure 24. The plaquettes of the honeycomb lattice in Figure 24 are colored in accordance with the colors of the cubes of the cubic lattice. We note that dynamical variables on the honeycomb lattice are simple objects, simple 1-morphisms, and basis 2-morphisms living on plaquettes, edges, and vertices.[35]

Based on the above definition of a time slice, we define the transfer matrix of the 3d height model on $\Lambda$. The transfer matrix $\hat{T}$ is a linear map from the state space on a time slice at $t = 0$ to the state space on another time slice at $t = 1$. Physically, this linear map represents the imaginary time evolution from $t = 0$ to $t = 1$. The state space on a time slice is spanned by possible configurations of dynamical variables on the honeycomb lattice. Specifically, the state space $\mathcal{H}$ is given by $\mathcal{H} = \text{Span}\{|\Gamma\rangle\}$, where $|\Gamma\rangle$ denotes a state corresponding to a configuration $\Gamma$ on the honeycomb lattice. Pictorially, we will often write $|\Gamma\rangle$ as

$$|\Gamma\rangle = \left| \begin{array}{c} \Gamma_8 \\ \Gamma_6 \quad \Gamma_7 \\ \Gamma_4 \\ \Gamma_2 \quad \Gamma_3 \\ \Gamma_1 \end{array} \right\rangle, \tag{4.1}$$

where we omitted labels on the edges and vertices on the right-hand side in order to avoid cluttering the notation. The inner product of states in $\mathcal{H}$ is defined by $\langle \Gamma' | \Gamma \rangle = \delta_{\Gamma, \Gamma'}$.

If we choose the initial time slice at $t = 0$ and the final time slice at $t = 1$ as shown in Figure 23, the transfer matrix $\hat{T}$ is factorized into the product of three linear maps

$$\hat{T} = \hat{T}_{\text{yellow}} \hat{T}_{\text{green}} \hat{T}_{\text{blue}}, \tag{4.2}$$

---

[35]Equivalently, simple objects, simple 1-morphisms, and basis 2-morphisms are living on vertices, edges, and plaquettes of the triangular lattice.

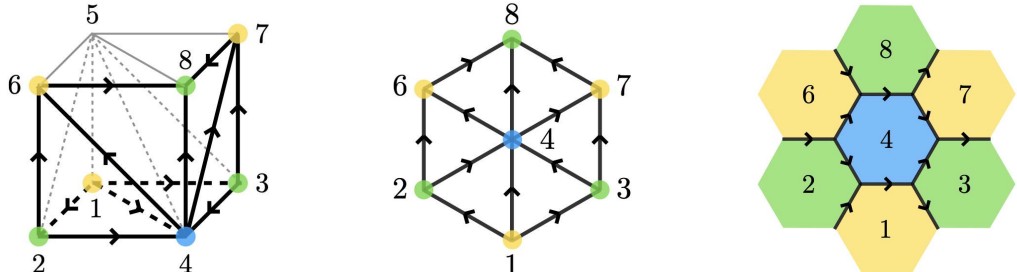

Figure 24: A time slice on the triangulated cubic lattice (left) becomes a triangular lattice (middle), which is related to a honeycomb lattice (right) by the Poincaré duality. The color of a vertex $v$ of the cubic lattice represents the color of the cube whose initial vertex is $v$. A vertex of the triangular lattice has the same color as the color of the corresponding vertex of the cubic lattice, while a plaquette of the honeycomb lattice has the same color as the color of its dual vertex of the triangular lattice.

where $\hat{T}_{\text{blue}}$, $\hat{T}_{\text{green}}$, and $\hat{T}_{\text{yellow}}$ are the imaginary time evolutions on the blue plaquettes, green plaquettes, and yellow plaquettes respectively. More specifically, the linear map $\hat{T}_{\text{blue}}$ is given by the product of local transfer matrices on the blue plaquettes. The matrix element of the local transfer matrix $\hat{T}_p$ on a plaquette $p$ is defined by the Boltzmann weight on the corresponding cube $c$ with the dynamical variables inside $c$ integrated out, namely,

$$\left\langle \begin{array}{c} \Gamma_8 \\ \Gamma_6 \quad \Gamma_7 \\ \Gamma_5 \\ \Gamma_2 \quad \Gamma_3 \\ \Gamma_1 \end{array} \middle| \hat{T}_p \middle| \begin{array}{c} \Gamma_8 \\ \Gamma_6 \quad \Gamma_7 \\ \Gamma_4 \\ \Gamma_2 \quad \Gamma_3 \\ \Gamma_1 \end{array} \right\rangle = \sum_{\Gamma_{\text{int}}^c} W_c(\Gamma; c), \tag{4.3}$$

where $c$ is a cube whose initial vertex is dual to the plaquette $p$ and $\Gamma_{\text{int}}^c$ is the collection of dynamical variables inside $c$. We note that the local transfer matrices on blue plaquettes commute with each other because any two plaquettes of the same color are not adjacent to each other. Hence, the product of these transfer matrices is defined unambiguously. The other two linear maps $\hat{T}_{\text{green}}$ and $\hat{T}_{\text{yellow}}$ in eq. (4.2) are also defined by the products of local transfer matrices on the green plaquettes and the yellow plaquettes respectively. Therefore, we have

$$\hat{T} = \prod_{\text{yellow plaquettes } p_y} \hat{T}_{p_y} \prod_{\text{green plaquettes } p_g} \hat{T}_{p_g} \prod_{\text{blue plaquettes } p_b} \hat{T}_{p_b}. \tag{4.4}$$

The partition function (3.3) of the 3d height model can be written in terms of the above transfer matrix as $Z(A) = \text{Tr}_{\mathcal{H}}(\hat{T}^N)$, where $N$ is the number of lattice sites in the time direction.

The transfer matrix formalism of the 3d height model enables us to write down the Hamiltonian of the corresponding 2+1d quantum model on a honeycomb lattice. Specifically, we define the Hamiltonian of the 2+1d quantum lattice model by

$$H = - \sum_{\text{plaquettes } p} \hat{T}_p, \tag{4.5}$$

where $\hat{T}_p$ is the local transfer matrix on a plaquette $p$ defined by eq. (4.3). However, the above 2+1d model is not precisely the quantum counterpart of the 3d height model. This

is because the imaginary time evolution $e^{-\epsilon H}$ on the state space $\mathcal{H}$ does not become the transfer matrix of the 3d height model even when $\epsilon \ll 1$. In other words, the transfer matrix $\hat{T}$ of the 3d height model cannot be expanded as $\hat{T} = \mathrm{id}_{\mathcal{H}} - \epsilon H + \mathcal{O}(\epsilon^2)$ due to the fact that the state space $\mathcal{H}$ contains a lot of states that are redundant in the description of the 3d height model.

The appropriate quantum counterpart of the 3d height model is obtained by restricting the state space of the above 2+1d model to a specific subspace of $\mathcal{H}$. To see this, we first notice that the partition function of the 3d height model can also be written as

$$Z(A) = \mathrm{Tr}_{\mathcal{H}} \left( (\hat{T}_0 \hat{T} \hat{T}_0)^N \right) = \mathrm{Tr}_{\mathcal{H}_0}(\hat{T}^N), \tag{4.6}$$

where $\hat{T}_0$ is the transfer matrix for the trivial weight $A(\mathcal{F}_{\mathrm{int}}^c) = \delta_{\mathcal{F}_{\mathrm{int}}^c, 1}$ and $\mathcal{H}_0 := \hat{T}_0 \mathcal{H}$ is the image of $\hat{T}_0$. Here, the trivial weight means that $A(\mathcal{F}_{\mathrm{int}}^c)$ is one if $\mathcal{F}_{\mathrm{int}}^c$ is a trivial coloring and zero otherwise. We note that $\hat{T}_0$ is a projector, that is, it satisfies $\hat{T}_0^2 = \hat{T}_0$. This is because the Dirichlet boundary decorated by the trivial coloring is topological in the imaginary time direction. More explicitly, $\hat{T}_0$ can be written as the product of local commuting projectors as in eq. (4.21), which makes it clear that $\hat{T}_0$ is a projector. The first equality of eq. (4.6) follows from the relation $\hat{T} = \hat{T}_0 \hat{T} \hat{T}_0$, which is an immediate consequence of the fact that $\hat{T}_0$ is the transfer matrix for the trivial weight. The second equality of eq. (4.6) follows from the definition of $\mathcal{H}_0$. Equation (4.6) motivates us to consider a 2+1d quantum lattice model whose state space on the honeycomb lattice is $\mathcal{H}_0$ rather than $\mathcal{H}$. The Hamiltonian of this model is given by eq. (4.5), where the domain of the Hamiltonian is now restricted to $\mathcal{H}_0$. The restriction of the state space to $\mathcal{H}_0$ makes sense because the Hamiltonian (4.5) does not mix states in $\mathcal{H}_0$ and those in the kernel $\ker(\hat{T}_0)$ of $\hat{T}_0$ due to the equality $\hat{T} = \hat{T}_0 \hat{T} \hat{T}_0$. The 2+1d quantum lattice model on $\mathcal{H}_0$ is precisely the quantum counterpart of the 3d height model. Indeed, if we take the weight $A(\mathcal{F}_{\mathrm{int}}^c)$ of the 3d height model to be slightly off from the trivial weight, i.e., $\delta_{\mathcal{F}_{\mathrm{int}}^c, 1} + \epsilon A(\mathcal{F}_{\mathrm{int}}^c)$ for $\epsilon \ll 1$, the transfer matrix $\hat{T}$ can be expanded in $\epsilon$ as $\hat{T} = \hat{T}_0 - \epsilon H \hat{T}_0 + \mathcal{O}(\epsilon^2)$,[36] which reduces to $\mathrm{id}_{\mathcal{H}_0} - \epsilon H + \mathcal{O}(\epsilon^2)$ on $\mathcal{H}_0$. Namely, our 2+1d lattice model whose state space is $\mathcal{H}_0$ can be obtained by taking the anisotropic limit of the 3d height model. Here, we emphasize that the point in passing to the smaller state space $\mathcal{H}_0$ is to find the completely anisotropic limit where the transfer matrix becomes the identity, around which we can expand the transfer matrix.

The introduction of $\mathcal{H}_0$ is also motivated by the 3+1d perspective. Specifically, the state space of the DR theory on a time slice that is perpendicular to the boundaries depicted in Figure 18, is represented by $\mathcal{H}_0$, not $\mathcal{H}$. This can be understood by noting that $\hat{T}_0$ is the transfer matrix when the decoration remains invariant under time translation, and such a transfer matrix should be evaluated as the identity operator on the state space.

There is also another way to see the reduction of the state space from a 3+1d point of view. As we will discuss in Section 4.3, $\hat{T}_0$ is the projection onto the eigenspace of a generalized Levin-Wen plaquette operator. This insight implies that $\mathcal{H}$ corresponds to the state space for configurations where a bunch of hollow cylinders connects the two boundaries of the 3+1d slab in Figure 18. Each cylinder terminates at the center of a plaquette on the right boundary and extends horizontally in the figure, and inside it, there is the trivial phase. The projector $\hat{T}_0$ is responsible for closing these holes, cf. the original discussion by Levin and Wen [125, Appendix C].

The discrepancy between the naive coloring space $\mathcal{H}$ and the space $\mathcal{H}_0$ stems from the difference in the homotopy type of the honeycomb lattice, on which the coloring is defined,

---

[36]We note that $H\hat{T}_0 = \hat{T}_0 H \hat{T}_0$ because $\hat{T}$ and $\hat{T}_0$ commute with each other due to $\hat{T} = \hat{T}_0 \hat{T} \hat{T}_0$.

and the homotopy type of the continuum space on which the DR theory is defined. The former has almost as many generators as plaquettes, while the latter depends only on the global shape of the space (e.g. either torus or open disk). This discrepancy does not occur in 1+1d open anyon chain.

We note that the derivation of the above 2+1d quantum lattice model from the 3d height model is parallel to the derivation of the 1+1d anyon chain model from the 2d AFM height model elaborated on in [33]. Thus, we can think of our lattice model as a 2+1d analogue of the anyon chain model. Indeed, as we will see in Section 4.3, our 2+1d model admits a graphical representation analogous to the anyon chain model. We call these 2+1d lattice models fusion surface models.

As we will discuss in Section 4.4, the 2+1d fusion surface model has an exact fusion 2-category symmetry described by the input fusion 2-category $\mathcal{C}$. Equivalently, the 2+1d quantum lattice model defined by the same form of the Hamiltonian (4.5) acting on a larger state space $\mathcal{H}$ has a fusion 2-category symmetry $\mathcal{C}$ only on its subspace $\mathcal{H}_0 \subset \mathcal{H}$.[37] The existence of this fusion 2-category symmetry is guaranteed by the symmetry TFT construction depicted in Figure 18.

## 4.2 Unitarity of the model

In this subsection, we spell out the condition for the Hamiltonian (4.5) to be Hermitian under several assumptions on the input fusion 2-category $\mathcal{C}$. Let us first list the assumptions that we make. The first assumption is that the set of representatives of the connected components of simple objects is closed under taking the dual up to isomorphism. Namely, for the representative $X$ of every connected component, there is a connected component whose representative $Y$ is isomorphic to the dual object $X^{\#}$. This isomorphism is assumed to preserve the quantum dimension and the 10-j symbol. The precise meaning of this assumption will become clear in a later computation. Similarly, for every representative $x$ of simple 1-morphisms in $\mathrm{Hom}_{\mathcal{C}}(X, Y)$, there is a representative $y$ of simple 1-morphisms in $\mathrm{Hom}_{\mathcal{C}}(Y, X)$ that is isomorphic to $x^*$, and we assume that this isomorphism preserves the 10-j symbol.[38] We also make an assumption that the quantum dimensions of the representatives of simple objects and simple 1-morphisms are positive real numbers.[39] Finally, we assume that the 10-j symbol has the properties that we call the reflection positivity and the 4-simplex symmetry. The reflection positivity of the 10-j symbol $z_{\epsilon}(\Gamma; [01234])$ is the property that flipping the orientation of a 4-simplex [01234] amounts to taking the complex conjugation of the 10-j symbol:[40]

$$z_{-\epsilon}(\Gamma; [01234]) = z_{\epsilon}(\Gamma; [01234])^*. \tag{4.7}$$

The 4-simplex symmetry of the 10-j symbol $z_{\epsilon}(\Gamma; [01234])$ is the invariance under any permutation $\sigma \in S^5$ of vertices of a 4-simplex [01234]:

$$z_{\epsilon}(\Gamma; [01234]) = z_{\epsilon \cdot \mathrm{sgn}(\sigma)}(\Gamma; [\sigma(0)\sigma(1)\sigma(2)\sigma(3)\sigma(4)]). \tag{4.8}$$

---

[37]This kind of symmetry is called exact emergent symmetry in [161].

[38]This assumption particularly implies that the Frobenius-Schur indicator of a self-dual simple 1-morphism is trivial.

[39]The quantum dimension of a simple object is multiplied by $\lambda^2$ if we stack an invertible 2d TFT on top of it, where $\lambda^2$ is the partition function of the invertible 2d TFT on a sphere. We note that $\lambda$ has to be real and hence $\lambda^2$ is positive if this invertible 2d TFT is reflection positive [162].

[40]The term "reflection positivity" originates from an analogy to the property of a reflection positive quantum field theory: the partition function of a reflection positive quantum field theory becomes its complex conjugate if the orientation of the underlying spacetime is reversed.

The signature $\text{sgn}(\sigma)$ of a permutation $\sigma$ is $+$ if $\sigma$ is an even permutation and it is $-$ if $\sigma$ is an odd permutation.[41]

When the permutation $\sigma$ is non-trivial, the right-hand side of eq. (4.8) involves objects and morphisms that are dual to those on the left-hand side. Let us illustrate this point by considering the simplest example where $\sigma = (01)$ is the transposition of 0 and 1. In this case, the 4-simplex symmetry (4.8) reduces to $z_\epsilon(\Gamma; [01234]) = z_{-\epsilon}(\Gamma; [10234])$. The right-hand side of this equation involves a simple object $\Gamma_{10}$, whereas the left-hand side involves another simple object $\Gamma_{01}$. These simple objects are supposed to be dual to each other, i.e., we have $\Gamma_{10} = \Gamma_{01}^{\#}$. Similarly, the right-hand side $z_{-\epsilon}(\Gamma; [10234])$ involves a simple 1-morphism $\Gamma_{10j} : \Gamma_{10} \square \Gamma_{0j} \to \Gamma_{1j}$ for $2 \leq j \leq 4$, whereas the left-hand side $z_\epsilon(\Gamma; [01234])$ involves another simple 1-morphism $\Gamma_{01j} : \Gamma_{01} \square \Gamma_{1j} \to \Gamma_{0j}$. These simple 1-morphisms are related to each other by an appropriate duality that contains both the object-level duality and the morphism-level duality. Specifically, the relation between $\Gamma_{01j}$ and $\Gamma_{10j}$ is expressed as

$$ \tag{4.9} $$

where $\Gamma_{01j}^* : \Gamma_{0j} \to \Gamma_{01} \square \Gamma_{1j}$ is the morphism-level dual of $\Gamma_{01j}$ and $(\Gamma_{01j}^*)^{\#} : \Gamma_{1j}^{\#} \square \Gamma_{01}^{\#} \to \Gamma_{0j}^{\#}$ is the object-level dual of $\Gamma_{01j}^*$. The relation between the basis 2-morphisms on the left-hand side $Z_\epsilon(\Gamma; [01234])$ and those on the right-hand side $z_{-\epsilon}(\Gamma; [10234])$ is also given in a similar way. The 4-simplex symmetry (4.8) implies that the 10-j symbol on a 4-simplex does not depend on the choice of a branching structure on it. This is a natural generalization of the tetrahedral symmetry of the 6-j symbol of a fusion 1-category [163].

Let us now derive the condition for the Hermiticity of the Hamiltonian (4.5) based on the above assumptions. We first write down the matrix element of the local Hamiltonian $\hat{T}_p$ explicitly as follows:

$$ \left\langle \begin{array}{c} \Gamma_8 \\ \Gamma_6 \; \Gamma_7 \\ \Gamma_5 \\ \Gamma_2 \; \Gamma_3 \\ \Gamma_1 \end{array} \middle| \hat{T}_p \middle| \begin{array}{c} \Gamma_8 \\ \Gamma_6 \; \Gamma_7 \\ \Gamma_4 \\ \Gamma_2 \; \Gamma_3 \\ \Gamma_1 \end{array} \right\rangle $$

$$ = \sum_{\Gamma_{45}} \sum_{\Gamma_{145},\cdots,\Gamma_{458}} \sum_{\mathcal{F}_{45}} \sum_{\mathcal{F}_{145},\cdots,\mathcal{F}_{458}} \sum_{\mathcal{F}_{1245},\cdots,\mathcal{F}_{4578}} A(\mathcal{F}_{45}; \mathcal{F}_{145}, \cdots, \mathcal{F}_{458}; \mathcal{F}_{1245}, \cdots, \mathcal{F}_{4578}) $$

$$ \dim(\Gamma_{45}) \sqrt{\prod_{j=4,5} \frac{\dim(\Gamma_{1j})\dim(\Gamma_{2j})\dim(\Gamma_{3j})\dim(\Gamma_{j6})\dim(\Gamma_{j7})\dim(\Gamma_{j8})}{\text{Dim}(\Gamma_j)}} $$

$$ z_\epsilon(\Gamma, \mathcal{F}; \text{pt} * [1245]) z_\epsilon(\Gamma, \mathcal{F}; \text{pt} * [2456]) z_\epsilon(\Gamma, \mathcal{F}; \text{pt} * [4568]) $$

$$ z_{-\epsilon}(\Gamma, \mathcal{F}; \text{pt} * [1345]) z_{-\epsilon}(\Gamma, \mathcal{F}; \text{pt} * [3457]) z_{-\epsilon}(\Gamma, \mathcal{F}; \text{pt} * [4578]). $$

$$ \tag{4.10} $$

---

[41]We expect that these conditions, e.g., the triviality of the Frobenius-Schur indicators of 1-morphisms, can be relaxed to the axioms of what we should call a unitary fusion 2-category. In the more general cases, the Hermiticity condition (4.12) should be modified to include, e.g., the Frobenius-Schur indicators: see Section 5.1. We do not explore the most general conditions in this paper.

The summation on the right-hand side is taken over the representatives of simple objects, simple 1-morphisms, and basis 2-morphisms. Due to the assumptions, the matrix element of the Hermitian conjugate of $\hat{T}_p$ can be computed as

$$
\left\langle \begin{array}{c}\Gamma_8 \\ \Gamma_6 \searrow \Gamma_7 \\ \rightarrow \Gamma_5 \rightarrow \\ \Gamma_2 \Gamma_3 \\ \Gamma_1\end{array} \middle| \hat{T}_p^\dagger \middle| \begin{array}{c}\Gamma_8 \\ \Gamma_6 \Gamma_7 \\ \rightarrow \Gamma_4 \rightarrow \\ \Gamma_2 \Gamma_3 \\ \Gamma_1\end{array} \right\rangle = \left\langle \begin{array}{c}\Gamma_8 \\ \Gamma_6 \Gamma_7 \\ \rightarrow \Gamma_4 \rightarrow \\ \Gamma_2 \Gamma_3 \\ \Gamma_1\end{array} \middle| \hat{T}_p \middle| \begin{array}{c}\Gamma_8 \\ \Gamma_6 \Gamma_7 \\ \rightarrow \Gamma_5 \rightarrow \\ \Gamma_2 \Gamma_3 \\ \Gamma_1\end{array} \right\rangle^*
$$

$$
= \sum_{\Gamma_{54}} \sum_{\Gamma_{154},\cdots,\Gamma_{548}} \sum_{\mathcal{F}_{54}} \sum_{\mathcal{F}_{154},\cdots,\mathcal{F}_{548}} \sum_{\mathcal{F}_{1254},\cdots,\mathcal{F}_{5478}} A(\mathcal{F}_{54};\mathcal{F}_{154},\cdots,\mathcal{F}_{548};\mathcal{F}_{1254},\cdots,\mathcal{F}_{5478})^*
$$

$$
\dim(\Gamma_{45})\sqrt{\prod_{j=4,5}\frac{\dim(\Gamma_{1j})\dim(\Gamma_{2j})\dim(\Gamma_{3j})\dim(\Gamma_{j6})\dim(\Gamma_{j7})\dim(\Gamma_{j8})}{\mathrm{Dim}(\Gamma_j)}}
$$

$$
z_\epsilon(\Gamma,\mathcal{F};\mathrm{pt}*[1245])z_\epsilon(\Gamma,\mathcal{F};\mathrm{pt}*[2456])z_\epsilon(\Gamma,\mathcal{F};\mathrm{pt}*[4568])
$$

$$
z_{-\epsilon}(\Gamma,\mathcal{F};\mathrm{pt}*[1345])z_{-\epsilon}(\Gamma,\mathcal{F};\mathrm{pt}*[3457])z_{-\epsilon}(\Gamma,\mathcal{F};\mathrm{pt}*[4578]).
$$

(4.11)

The Hamiltonian (4.5) is Hermitian if and only if the above two quantities (4.10) and (4.11) agree with each other. We emphasize that $\Gamma_{45}, \mathcal{F}_{45}$, etc. involved in the 10-j symbols in eq. (4.11) are not representatives themselves in general but the appropriate duals of the representatives $\Gamma_{54}, \mathcal{F}_{54}$, etc. Although they are not representatives, they are isomorphic to representatives because the set of representatives is assumed to be closed under taking the dual up to isomorphism. Since we are assuming that these isomorphisms preserve the quantum dimension and 10-j symbol, we can identify the summands on the right-hand side of eq. (4.11) with those on the right-hand side of eq. (4.10). Therefore, the Hermiticity condition on the Hamiltonian (4.5) reduces to

$$
A(\mathcal{F}_{45};\mathcal{F}_{145},\cdots,\mathcal{F}_{458};\mathcal{F}_{1245},\cdots,\mathcal{F}_{4578}) = A(\mathcal{F}_{54};\mathcal{F}_{154},\cdots,\mathcal{F}_{548};\mathcal{F}_{1254},\cdots,\mathcal{F}_{5478})^*,
$$

(4.12)

where the arguments on the right-hand side are the representatives of the connected components of appropriate duals of the arguments on the left-hand side. The above equation can be written simply as $A(\mathcal{F}^c_{\mathrm{int}}) = A(\overline{\mathcal{F}}^c_{\mathrm{int}})^*$, where the bar represents the appropriate dual.[42]

### 4.3 Graphical representation

In this subsection, we give a graphical representation of the 2+1d fusion surface model that we obtained from the 3d height model. To begin with, we consider a graphical representation of a state in the larger state space $\mathcal{H}$. As we mentioned in Section 4.1, states in $\mathcal{H}$ are in one-to-one correspondence with possible configurations of dynamical variables on a honeycomb lattice. A configuration of dynamical variables is constrained by the monoidal structure of the input fusion 2-category $\mathcal{C}$. For example, a simple 1-morphism $\Gamma_{ij}$ on an edge $e_{ij}$ is constrained by the simple objects $\Gamma_i$ and $\Gamma_j$ on the adjacent plaquettes $p_i$ and $p_j$. More specifically, $\Gamma_{ij}$ has to be a simple 1-morphism from $\Gamma_i \square \mathcal{F}_{ij}$ to $\Gamma_j$, where $\mathcal{F}_{ij}$ is a simple object assigned to a 1-simplex $[ij]$ of the original 3d lattice $\Lambda$ that is dual to an edge $e_{ij}$ on the honeycomb lattice. We recall that $\mathcal{F}_{ij}$ is fixed due to eq. (3.6), meaning that $\mathcal{F}_{ij}$ is not dynamical. Similarly, a basis 2-morphism $\Gamma_{ijk}$ on a vertex $v_{ijk}$ at the junction of three edges $e_{ij}$, $e_{jk}$, and $e_{ik}$ must be a 2-morphism between

---

[42]In Section 5.1, we will see an example where the Hermiticity condition (4.12) is modified due to the non-trivial Frobenius-Schur indicator of a simple 1-morphism.

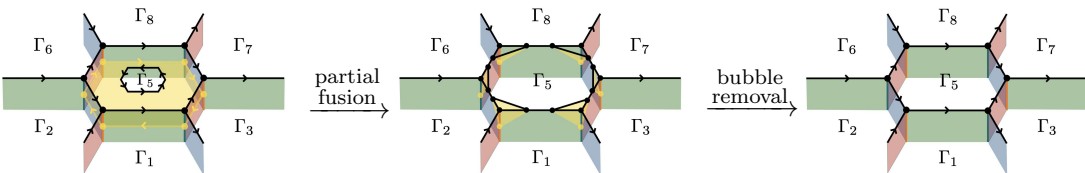

Figure 25: The diagrammatic representation of the Hamiltonian is evaluated by combining the partial fusion around edges and the bubble removal around vertices.

$\Gamma_{jk} \circ (\Gamma_{ij} \square \mathbf{1}_{\mathcal{F}_{jk}})$ and $\Gamma_{ik} \circ (\mathbf{1}_{\Gamma_i} \square \mathcal{F}_{ijk})$. The local constraints around all vertices combine dynamical variables on the honeycomb lattice into a single fusion diagram. Therefore, a state on the honeycomb lattice can be identified with a fusion diagram as follows:

$$\left| \begin{array}{c} \Gamma_8 \\ \Gamma_6 \quad \Gamma_7 \\ \Gamma_4 \\ \Gamma_2 \quad \Gamma_3 \\ \Gamma_1 \end{array} \right\rangle = \mathcal{N}_\Gamma \quad \text{(diagram)} . \tag{4.13}$$

Here, the left-hand side is an orthonormal basis of the state space $\mathcal{H}$ on the honeycomb lattice and $\mathcal{N}_\Gamma$ is a normalization factor defined by

$$\mathcal{N}_\Gamma = \sqrt{\prod_{\text{plaquettes}} \frac{1}{\text{Dim}(\Gamma_i)} \prod_{\text{edges}} \dim(\Gamma_{ij})} . \tag{4.14}$$

The action of the local Hamiltonian $\hat{T}_p$ on a state (4.13) is graphically expressed as

$$\hat{T}_p \quad \text{(diagram)} = \sum_{\Gamma_5} \sum_{\Gamma_{45}} \sum_{\mathcal{F}_{\text{int}}} A(\mathcal{F}_{\text{int}}) \frac{\dim(\Gamma_{45})}{\text{Dim}(\Gamma_5)} \quad \text{(diagram)} . \tag{4.15}$$

The yellow surface and the small white plaquette on the right-hand side are labeled by simple objects $\mathcal{F}_{45}$ and $\Gamma_5$ respectively. The edges and vertices are also labeled by simple 1-morphisms and basis 2-morphisms, although the labels are omitted in the above equation due to the lack of space. For example, the loop at the junction of three surfaces $\Gamma_4, \Gamma_5$, and $\mathcal{F}_{45}$ is labeled by a simple 1-morphism $\Gamma_{45} : \Gamma_4 \square \mathcal{F}_{45} \to \Gamma_5$. The other labels can also be deduced from the labels already specified in the above equation. The right-hand side of eq. (4.15) is evaluated in two steps as shown in Figure 25. In what follows, we show that the above graphical representation gives the correct matrix element (4.3) of the local Hamiltonian $\hat{T}_p$ by explicitly evaluating the fusion diagram step by step.

The first step of the evaluation is the partial fusion, which is represented by the following diagrammatic equality of 2-morphisms:

$$\text{(diagram)} = \sum_d \sum_{(\pi_i, \iota_i)} \text{(diagram)} = \sum_d \sum_{\alpha_i} \dim(d) \quad \text{(diagram)} . \tag{4.16}$$

The 2-morphisms $\pi_i$ and $\iota_i$ in the above equation are the projection and inclusion 2-morphisms, whereas $\alpha_i$ and $\overline{\alpha}_i$ are basis 2-morphisms. The defining property of the projection and inclusion 2-morphisms is that they are dual to each other and satisfy $\pi_i \cdot \iota_j = \delta_{ij}\mathbf{1}$. In particular, 2-morphisms $\pi_i$ and $\iota_i$ are proportional to basis 2-morphisms $\alpha_i$ and $\overline{\alpha}_i$. The proportionality constant can be figured out by comparing the trace of $\iota_i \cdot \pi_i$ with that of $\overline{\alpha}_i \cdot \alpha_i$. The trace of $\iota_i \cdot \pi_i$ is equal to the quantum dimension of the target 1-morphism $d$ of $\pi_i$, while the trace of $\overline{\alpha}_i \cdot \alpha_i$ is unity because $\alpha_i$ is normalized. Therefore, we have $\iota_i \cdot \pi_i = \dim(d)\overline{\alpha}_i \cdot \alpha_i$, which shows the second equality of eq. (4.16). We perform this partial fusion for all edges around the central plaquette labeled by $\Gamma_5$.

The second step of the evaluation is to remove the small bubbles that are localized around the vertices after we perform the partial fusion. As an example, we focus on the bubble at the left bottom vertex $v_{124}$. In order to remove the bubble, we first notice that the configuration of surfaces around a vertex $v_{124}$ can be identified with the left-hand side of the 10-j move (2.3) as follows:[43]

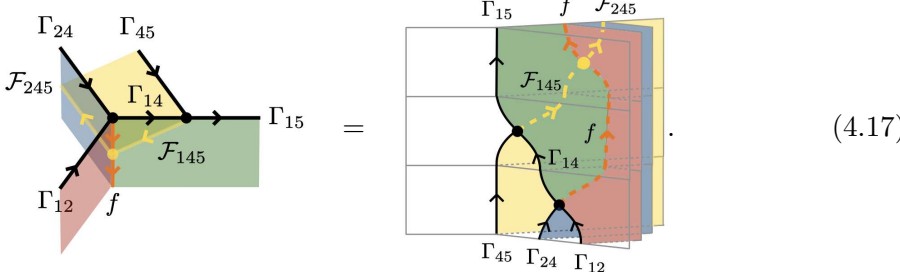

$$ (4.17) $$

This identification makes it clear that the 10-j move around a vertex $v_{124}$ deforms the fusion diagram as

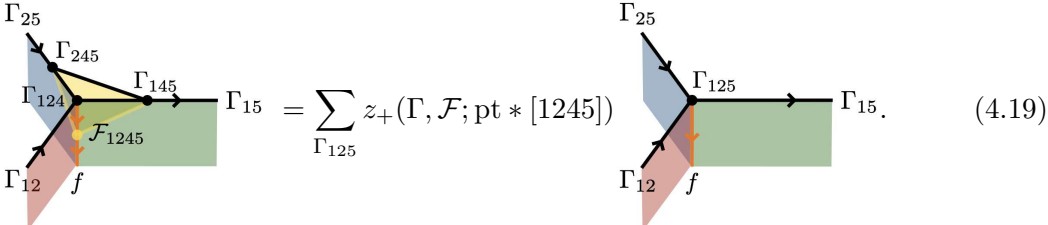

$$ (4.18) $$

We can now remove the bubble on the right-hand side by using the fact that the composition of $\Gamma'_{245}$ and $\Gamma_{245}$ is non-zero only when $\Gamma'_{25} = \Gamma_{25}$ and $\Gamma'_{245} = \overline{\Gamma}_{245}$. When non-zero, the above composite map is a 2-endomorphism of $\Gamma_{25}$, which is proportional to the identity 2-morphism because $\Gamma_{25}$ is simple. More specifically, we have $\Gamma'_{245} \cdot \Gamma_{245} = \delta_{\Gamma_{25},\Gamma'_{25}}\delta_{\overline{\Gamma}_{245},\Gamma'_{245}}\dim(\Gamma_{25})^{-1}\mathrm{id}_{\Gamma_{25}}$, which can be verified by computing the trace of both sides. Therefore, eq. (4.18) reduces to

$$ \begin{array}{ccc} \Gamma_{25} \\ \Gamma_{245} \\ \Gamma_{124} \quad \Gamma_{145} \\ \mathcal{F}_{1245} \quad \Gamma_{15} & = \displaystyle\sum_{\Gamma_{125}} z_+(\Gamma, \mathcal{F}; \mathrm{pt} * [1245]) & \Gamma_{25} \\ \Gamma_{12} \quad f & & \Gamma_{12} \quad f \end{array} \qquad (4.19) $$

Similar equations also hold for the other vertices.[44]

---

[43]Equation (4.17) involves the identification of 2-morphisms related by the duality.

[44]Precisely, the 10-j symbol $z_+$ on the right-hand side is replaced by $z_-$ depending on vertices.

Combining eqs. (4.13), (4.14), (4.15), (4.16), and (4.19) leads to eq. (4.10) with $\epsilon$ being $+$. Thus, we find that the 2+1d quantum lattice model defined on the larger state space $\mathcal{H}$ has a graphical representation (4.15). This graphical representation can further be simplified on the subspace $\mathcal{H}_0 \subset \mathcal{H}$ as follows:

$$\hat{T}_p \quad \overset{\Gamma_8}{\underset{\Gamma_1}{\Gamma_6 \ \Gamma_4 \ \Gamma_7}{\Gamma_2 \ \Gamma_3}} \quad = \sum_{\mathcal{F}_{\text{int}}} A(\mathcal{F}_{\text{int}}) \quad \overset{\Gamma_8}{\underset{\Gamma_1}{\Gamma_6 \ \mathcal{F}_{45} \ \Gamma_4 \ \Gamma_7}{\Gamma_2 \ \Gamma_3}}. \tag{4.20}$$

This is the graphical representation of the Hamiltonian of the 2+1d fusion surface model. We will show the above equation in the rest of this subsection.

Before we derive eq. (4.20), we first specify the subspace $\mathcal{H}_0 \subset \mathcal{H}$ in more detail. As alluded to in Section 4.1, the subspace $\mathcal{H}_0$ is defined as the image of the transfer matrix $\hat{T}_0$ for the trivial weight. The transfer matrix $\hat{T}_0$ is given by the product of local transfer matrices $\hat{B}_p$, namely,

$$\hat{T}_0 = \prod_{\text{plaquettes } p} \hat{B}_p, \tag{4.21}$$

where $\hat{B}_p$ is represented by the following diagrammatic equation:

$$\hat{B}_p \quad \overset{\Gamma_8}{\underset{\Gamma_1}{\Gamma_6 \ \Gamma_4 \ \Gamma_7}{\Gamma_2 \ \Gamma_3}} \quad = \sum_{\Gamma_{45} \in \text{End}(\Gamma_4)} \frac{\dim(\Gamma_{45})}{\text{Dim}(\Gamma_4)} \quad \overset{\Gamma_8}{\underset{\Gamma_1}{\Gamma_6 \ \Gamma_{45} \ \Gamma_4 \ \Gamma_7}{\Gamma_2 \ \Gamma_3}}. \tag{4.22}$$

We note that $\hat{B}_p$ is a local commuting projector, i.e., it satisfies $\hat{B}_p \hat{B}_{p'} = \hat{B}_{p'} \hat{B}_p$ and $\hat{B}_p^2 = \hat{B}_p$.[45] Therefore, the subspace $\mathcal{H}_0$ is spanned by the states satisfying $\hat{B}_p = 1$ for all the plaquettes:

$$\mathcal{H}_0 = \hat{T}_0 \mathcal{H} = \text{Span}\{|\Gamma\rangle \in \mathcal{H} \mid \hat{B}_p |\Gamma\rangle = |\Gamma\rangle, \forall p\}. \tag{4.23}$$

On this subspace, a contractible loop of $x \in \text{End}(\Gamma_4)$ on a plaquette acts as a scalar multiplication. This is because the loop operator $\hat{B}_p^x$ for a contractible loop of $x$ on a plaquette $p$ can be absorbed by the projector $\hat{B}_p$ as follows:

$$\hat{B}_p^x \hat{B}_p = \hat{B}_p^x \sum_{\Gamma_{45} \in \text{End}(\Gamma_4)} \frac{\dim(\Gamma_{45})}{\text{Dim}(\Gamma_4)} \hat{B}_p^{\Gamma_{45}} = \frac{\dim(x)}{\dim(\Gamma_4)} \hat{B}_p. \tag{4.24}$$

This equation implies that a contractible loop of $x$ can be shrunk at the expense of multiplying a scalar factor $\dim(x)/\dim(\Gamma_4)$, which is the quantum dimension of an object $x$ in a fusion 1-category $\text{End}(\Gamma_4)$.

On the subspace $\mathcal{H}_0$, we can also define the states whose plaquette variables are not representatives in $\pi_0 \mathcal{C}$. This is achieved by demanding that the contractible loop on a plaquette can be shrunk at the expense of multiplying a scalar factor. Specifically, such a

---

[45]The local commuting projector $\hat{B}_p$ is nothing but the plaquette term of the Levin-Wen model for the input fusion 1-category $\text{End}(\Gamma_4)$ [125].

state is defined by

$$\tag{4.25}$$

The left-hand side is well-defined on $\mathcal{H}_0$ because the right-hand side does not depend on the choice of $x \in \mathrm{Hom}(\Gamma'_4, \Gamma_4)$ when projected onto $\mathcal{H}_0$. Indeed, the composite of the loop operator $\hat{B}_p^x$ on the right-hand side and the projector $\hat{B}_p$ is independent of $x$:

$$\frac{\dim(\Gamma'_4)}{\dim(x)}\hat{B}_p\hat{B}_p^x = \sum_{y \in \mathrm{Hom}(\Gamma'_4, \Gamma_4)} \frac{\dim(y)}{\mathrm{Dim}(\Gamma_4)}\hat{B}_p^y. \tag{4.26}$$

Let us now show that eq. (4.15) reduces to eq. (4.20) on $\mathcal{H}_0$. To this end, we use the following expression for a simple 1-morphism $\Gamma_{45} : \Gamma_4 \square \mathcal{F}_{45} \to \Gamma_5$ on the right-hand side of eq. (4.15):

$$\Gamma_{45} \cong \Gamma'_{45} \circ P. \tag{4.27}$$

Here, $P$ is the projection 1-morphism from $\Gamma_4 \square \mathcal{F}_{45} \cong \boxplus \Gamma'_5$ to a fusion channel $\Gamma'_5$ and $\Gamma'_{45}$ is a simple 1-morphism from $\Gamma'_5$ to $\Gamma_5$. We note that the fusion channels are uniquely determined only up to isomorphism. Physically, isomorphic fusion channels differ by invertible 2d TFTs stacked to topological surfaces. We can and will always choose the fusion channels properly so that the dimension of the projection 1-morphism $P$ agrees with the dimension of its target $\Gamma'_5$, i.e., we have $\dim(P) = \dim(\Gamma'_5)$. This choice of the fusion channels in particular implies that simple 1-morphisms $\Gamma_{45}$ and $\Gamma'_{45}$ have the same dimension. By substituting eq. (4.27) into the right-hand side of eq. (4.15) and shrinking the loop of $\Gamma'_{45}$, we find

$$\tag{4.28}$$

Due to the equality $\dim(\mathrm{End}(\Gamma_5)) = \sum_{\Gamma'_{45} \in \mathrm{Hom}(\Gamma'_5, \Gamma_5)} \dim(\Gamma'_{45})^2 / \dim(\Gamma_5)\dim(\Gamma'_5)$, which was shown in [55], the above equation reduces to

$$\tag{4.29}$$

The right-hand side of the above equation can be written as

$$\tag{4.30}$$

To show this equation, we notice that the fusion of two surfaces $\Gamma_4$ and $\mathcal{F}_{45}$ satisfies

$$\boxed{\Gamma_4 \square \mathcal{F}_{45}} = \sum_{\Gamma'_5, P} c(P) \; \boxed{\begin{array}{c} \overset{\curvearrowleft}{\boxed{\Gamma'_5}} P \\ \Gamma_4 \square \mathcal{F}_{45} \end{array}} \tag{4.31}$$

for some coefficient $c(P) \in \mathbb{C}$. If we compose the both sides with the inclusion 1-morphism $I : \Gamma'_5 \to \boxplus \Gamma'_5 \cong \Gamma_4 \square \mathcal{F}_{45}$, the above equation reduces to

$$\boxed{\begin{array}{c} \overset{\curvearrowleft}{\boxed{\Gamma_4 \square \mathcal{F}_{45}}} I \\ \Gamma'_5 \end{array}} = c(P) \; \boxed{\quad \Gamma'_5 \quad}, \tag{4.32}$$

where the projection 1-morphism $P$ on the right-hand side is the dual of the inclusion 1-morphism $I$ on the left-hand side. One can extract the coefficient $c(P)$ by taking the trace of eq. (4.32) as $c(P) = \dim(I)/\dim(\Gamma'_5) = 1$, where the last equality follows from $\dim(I) = \dim(P) = \dim(\Gamma'_5)$. Thus, we find that eq. (4.30) holds,[46] which then implies eq. (4.20) due to eq. (4.29).

## 4.4 Fusion 2-category symmetry

### 4.4.1 General case

The graphical representation (4.20) makes it clear that the 2+1d fusion surface model has a fusion 2-category symmetry. The action of a fusion 2-category symmetry $\mathcal{C}$ is defined by the fusion of surface defects and/or line defects to the fusion diagram representing a state, see Figure 16 for a schematic picture of the symmetry action. The commutativity of the Hamiltonian and the symmetry action automatically follows from the coherence conditions for a fusion 2-category. In the following subsections, we study several simple examples of fusion 2-category symmetries to demonstrate that the fusion surface models actually have symmetries whose actions are defined in the above fashion.

Before proceeding, we emphasize that in general, the action of a fusion 2-category symmetry is well-defined only on the projected state space $\mathcal{H}_0$. This is because if we try to define the symmetry action by fusing a topological surface defect to a fusion diagram representing a state, we generically end up with a fusion diagram whose plaquette variables are not in the set of representatives of simple objects. Such a fusion diagram can be canonically identified with a state only on $\mathcal{H}_0$ as in eq. (4.25).[47]

### 4.4.2 Non-invertible 1-form symmetry

The fusion 2-category that describes a (potentially) non-invertible 1-form symmetry is the 2-category $\mathrm{Mod}(\mathcal{B})$ of $\mathcal{B}$-module categories,[48] where $\mathcal{B}$ is the ribbon 1-category of topological line defects.[49] The fusion 2-category $\mathrm{Mod}(\mathcal{B})$ has only one connected component

---

[46]The fact that $P$ is the projection 1-morphism from $\Gamma_4 \square \mathcal{F}_{45}$ to $\Gamma'_5$ is not sufficient to show eq. (4.30) because the left-hand side of eq. (4.30) depends on the choice of the fusion channels through an isomorphism $\Gamma_4 \square \mathcal{F}_{45} \cong \boxplus \Gamma'_5$, whereas the right-hand side does not. Our claim is that eq. (4.30) is satisfied when each fusion channel $\Gamma'_5$ is chosen so that $\dim(P) = \dim(\Gamma'_5)$.

[47]As we will see in Section 4.4.2, the action of line defects is well-defined also on $\mathcal{H}$ because the edge variables are always in the set of representatives.

[48]Here, a 1-form symmetry refers to a symmetry generated by codimension 2 topological defects, which may or may not be invertible. This symmetry reduces to an ordinary (group-like) 1-form symmetry when the topological defects are invertible. In this paper, non-invertible 1-form symmetries are also simply called 1-form symmetries.

[49]The fusion 2-category $\mathrm{Mod}(\mathcal{B})$ is spherical when $\mathcal{B}$ is a ribbon 1-category, see Example 2.3.5 of [55].

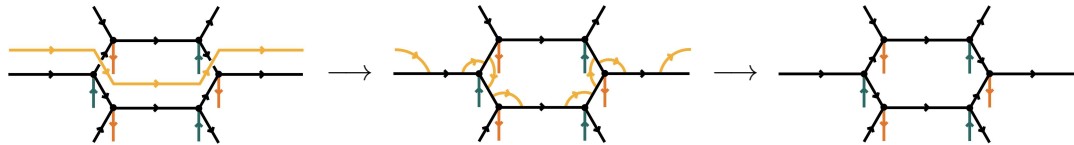

Figure 26: A (potentially) non-invertible 1-form symmetry $\mathcal{B}$ acts on states by the fusion of a topological line defect, which is written in light orange in the above figure. We use the $F$-symbols and the $R$-symbols of a ribbon 1-category $\mathcal{B}$ to fuse the line defect with the edges of a honeycomb lattice.

of simple objects, whose representative is chosen to be a unit object $I$, i.e., the regular $\mathcal{B}$-module. The endomorphism category of $I \in \mathrm{Mod}(\mathcal{B})$ is equivalent to $\mathcal{B}$. In a general fusion 2-category $\mathcal{C}$, the endomorphism category $\mathrm{End}_{\mathcal{C}}(I)$ describes the 1-form part of the whole symmetry. The action of the 1-form part of a general fusion 2-category symmetry $\mathcal{C}$ can be defined in the same way as the action of $\mathrm{Mod}(\mathcal{B})$ symmetry that we will discuss below. In this subsection, we will focus on the case where $\mathcal{C} = \mathrm{Mod}(\mathcal{B})$ for simplicity.

Since the connected component of simple objects of $\mathrm{Mod}(\mathcal{B})$ is unique, we do not have dynamical variables on the plaquettes of a honeycomb lattice. Therefore, the dynamical variables of the model are living only on the edges and vertices. These dynamical variables are labeled by simple objects and basis morphisms of a ribbon 1-category $\mathcal{B}$.

The diagrammatic representation (4.15) of the Hamiltonian acting on the larger state space $\mathcal{H}$ is given by

$$\hat{T}_p \;\; \begin{matrix} \text{(diagram)} \end{matrix} \;\; = \sum_{\Gamma_{45}} \sum_{\mathcal{F}_{\mathrm{int}}} A(\mathcal{F}_{\mathrm{int}}) \frac{\dim(\Gamma_{45})}{\mathcal{D}} \;\; \begin{matrix} \text{(diagram)} \end{matrix} \;\;, \quad (4.33)$$

where $\mathcal{D}$ is the total dimension of a ribbon 1-category $\mathcal{B}$. The right-hand side is evaluated by fusing the loops of $\Gamma_{45}$ and $\mathcal{F}_{\mathrm{int}} = \{\mathcal{F}_{145}, \cdots, \mathcal{F}_{458}\}$ to the nearby edges. The action of the 1-form symmetry is defined by the fusion of topological line defects to the honeycomb lattice as shown in Figure 26. The commutativity of the Hamiltonian and the symmetry action follows from the coherence conditions (i.e., the pentagon and hexagon equations) of a ribbon category. It is straightforward to generalize the action of a line defect illustrated in Figure 26 to the action of a general defect network, which also clearly commutes with the Hamiltonian (4.33). In particular, we can explicitly define the action of condensation defects on the lattice [56, 68, 164].

We emphasize that the action of $\mathrm{Mod}(\mathcal{B})$ symmetry commutes with the Hamiltonian not only on $\mathcal{H}_0$ but also on $\mathcal{H}$. This might seem to imply that our 2+1d lattice model has a non-invertible 1-form symmetry $\mathrm{Mod}(\mathcal{B})$ on the entire state space $\mathcal{H}$. However, the symmetry on $\mathcal{H}$ is not the usual 1-form symmetry because the action of a symmetry operator on a contractible loop is non-trivial. Such a symmetry on the lattice is called a 1-symmetry rather than 1-form symmetry in the literature [58, 136]. Therefore, our 2+1d lattice model has a 1-symmetry on the entire state space $\mathcal{H}$, which reduces to a 1-form symmetry $\mathrm{Mod}(\mathcal{B})$ on $\mathcal{H}_0$.

We note that the Hamiltonian (4.33) is factorized into the product of two commuting operators as

$$\hat{T}_p = \hat{B}_p \hat{T}'_p = \hat{T}'_p B_p, \quad (4.34)$$

where $\hat{B}_p$ and $\hat{T}'_p$ are defined by

$$\hat{B}_p \quad \cdots \quad = \sum_{\Gamma_{45}} \frac{\dim(\Gamma_{45})}{\mathcal{D}} \quad \cdots \quad , \tag{4.35}$$

$$\hat{T}'_p \quad \cdots \quad = \sum_{\mathcal{F}_{\text{int}}} A(\mathcal{F}_{\text{int}}) \quad \cdots \quad . \tag{4.36}$$

Eq. (4.35) can be regarded as a special case of eq. (4.36). Both of the above operators preserve the non-invertible 1-symmetry on $\mathcal{H}$. In particular, a new Hamiltonian defined by $H' = -\sum_p \hat{T}'_p$ also possesses the same non-invertible 1-symmetry.

### 4.4.3 Anomalous finite group symmetry

We consider the case of a finite group symmetry $G$ with an anomaly $[\omega] \in H^4(G, \mathrm{U}(1))$. A fusion 2-category $2\,\mathrm{Vec}_G^\omega$ describing an anomalous finite group symmetry $G$ consists of simple objects labeled by group elements. The 10-j symbol is given by a 4-cocycle $\omega$ as $z_+(\Gamma; [ijklm]) = \omega(\Gamma_{ij}, \Gamma_{jk}, \Gamma_{kl}, \Gamma_{lm})$. Since $2\,\mathrm{Vec}_G^\omega$ does not have non-trivial 1-morphisms and 2-morphisms in the sense that $\mathrm{Hom}_{2\mathrm{Vec}_G^\omega}(g, h) \cong \delta_{g,h}\mathrm{Vec}$ as a 1-category, we do not have dynamical variables on the edges and vertices of a honeycomb lattice. Thus, the dynamical variables are living only on the plaquettes. The dynamical variable on each plaquette takes values in $G$.

In order to obtain a non-trivial model with an anomalous finite group symmetry $G$, we need to choose objects $\rho, \sigma$, and $\lambda$ in eq. (3.6) to be non-simple.[50] In the following, we choose $\rho, \sigma$, and $\lambda$ to be the sum of all simple objects, i.e., we have

$$\rho = \sigma = \lambda = \bigboxplus_{g \in G} g. \tag{4.37}$$

In this case, there are effectively no constraints on the configuration of dynamical variables on the plaquettes. More specifically, the state space $\mathcal{H}$ of the model is given by the tensor product of local Hilbert spaces $\mathbb{C}^{|G|}$ on the plaquettes of the honeycomb lattice. We note that there is no difference between $\mathcal{H}$ and $\mathcal{H}_0$ for this example because the local commuting projector $\hat{B}_p$ on each plaquette is the identity operator due to the absence of non-trivial 1-morphisms.

The matrix element (4.10) of the local Hamiltonian $\hat{T}_p$ is given by

$$\left\langle \begin{array}{c} g_8 \\ g_6 \quad g_7 \\ g_5 \\ g_2 \quad g_3 \\ g_1 \end{array} \right| \hat{T}_p \left| \begin{array}{c} g_8 \\ g_6 \quad g_7 \\ g_4 \\ g_2 \quad g_3 \\ g_1 \end{array} \right\rangle$$

$$= A(g_4^{-1} g_5) \frac{\omega(g_1, g_1^{-1} g_2, g_2^{-1} g_4, g_4^{-1} g_5)\omega(g_2, g_2^{-1} g_4, g_4^{-1} g_5, g_5^{-1} g_6)\omega(g_4, g_4^{-1} g_5, g_5^{-1} g_6, g_6^{-1} g_8)}{\omega(g_1, g_1^{-1} g_3, g_3^{-1} g_4, g_4^{-1} g_5)\omega(g_3, g_3^{-1} g_4, g_4^{-1} g_5, g_5^{-1} g_7)\omega(g_4, g_4^{-1} g_5, g_5^{-1} g_7, g_7^{-1} g_8)}. \tag{4.38}$$

---

[50] Objects $\rho, \sigma$, and $\lambda$ were originally supposed to be simple in eq. (3.6). However, the diagrammatic representation of the model given in Section 4.3 enables us to generalize them to non-simple objects straightforwardly.

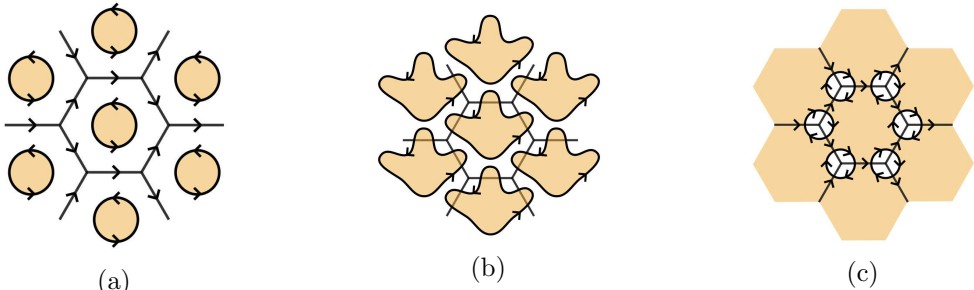

| (a) | (b) | (c) |

Figure 27: We fuse a surface defect to the honeycomb lattice from above by combining (a) the partial fusion, (b) the deformation of the defect, and (c) the bubble removal. The orange region in the above figures represents the region where the surface defect is already fused, while the white region represents the region where the surface defect is not fused yet.

The diagrammatic representation (4.20) of the above Hamiltonian clearly shows that this model has an anomalous finite group symmetry whose action is defined by fusing topological surface defects to the honeycomb lattice from above.

Let us explicitly compute the action of a finite group symmetry $G$ with anomaly $\omega$.[51] To this end, we precisely define the process of fusing a surface defect labeled by $g \in G$ to the honeycomb lattice from above. As illustrated in Figure 27a, instead of performing the fusion at one time, we first fuse a surface defect to the honeycomb lattice only inside each plaquette. We then slightly deform the surface defect as shown in Figure 27b and perform the partial fusion of the defect around the vertices so that the defect looks like Figure 27c. Removing the small bubbles in Figure 27c completes the fusion of a surface defect. As we discussed in Section 4.3, removing a bubble at a vertex amounts to multiplying the 10-j symbol. More specifically, we have

$$
\begin{aligned}
&\includegraphics{eq439a} = \omega(g, g_i, g_i^{-1}g_j, g_j^{-1}g_k)\ \includegraphics{eq439b}, \\
&\includegraphics{eq439c} = \omega(g, g_i, g_i^{-1}g_j, g_j^{-1}g_k)^*\ \includegraphics{eq439d}.
\end{aligned}
\tag{4.39}
$$

Therefore, the action of $g \in G$ on a general state $|\{g_i\}\rangle$ can be written as

$$
\hat{U}_g \left|\{g_i\}\right\rangle = \prod \omega(g, g_i, g_i^{-1}g_j, g_j^{-1}g_k) \prod \omega(g, g_i, g_i^{-1}g_j, g_j^{-1}g_k)^* \left|\{gg_i\}\right\rangle,
\tag{4.40}
$$

where the first and the second products on the right-hand side are taken over vertices shown in the first and the second equalities in eq. (4.39). A straightforward calculation shows that the symmetry action (4.40) commutes with the Hamiltonian (4.38) due to the cocycle condition on $\omega$. We note that the anomalous finite group symmetry of the Hamiltonian (4.38) is preserved even if the weight $A(g_4^{-1}g_5)$ also depends on other variables of the form $g_i^{-1}g_j$. In particular, when $G$ is anomaly-free, our model reduces to the $G$-symmetric models discussed in [113].

---

[51]See [165] for the action of an anomalous finite group symmetry in general dimensions.

# 5 Examples

In this section, we discuss several examples of the fusion surface model and its variants. In sections 5.1 and 5.2, we consider the 2+1d lattice models only with 1-symmetries. The 1-symmetries are present on the larger state space $\mathcal{H}$ as we observed in Section 4.4.2. As such, we consider the 2+1d lattice models defined on $\mathcal{H}$ without projecting to the subspace $\mathcal{H}_0$. On the other hand, in Section 5.3, we consider the 2+1d lattice models with general fusion 2-category symmetries. Since such symmetries are present only on the projected subspace $\mathcal{H}_0$, we need to consider the fusion surface models whose state space is $\mathcal{H}_0$ rather than $\mathcal{H}$. In the following, we use 1-symmetry and 1-form symmetry interchangeably when no confusion can arise. For the 2+1d models with 0-form anomalous finite group symmetries, see Section 4.4.3.

## 5.1 Lattice models with anomalous invertible 1-form symmetries

Let $A$ be a finite abelian group. Anomalies of an invertible 1-form symmetry $A$ are characterized by the $F$-symbols and $R$-symbols defined by the following equations:

$$
\begin{array}{c}
\vcenter{\hbox{\includegraphics{}}}
\end{array}
= F(a,b,c)
\begin{array}{c}
\vcenter{\hbox{\includegraphics{}}}
\end{array}
,
\qquad
\begin{array}{c}
\vcenter{\hbox{\includegraphics{}}}
\end{array}
= R(a,b)
\begin{array}{c}
\vcenter{\hbox{\includegraphics{}}}
\end{array}
. \tag{5.1}
$$

The quantum dimensions of invertible lines are all given by one, i.e., $\dim(a) = 1$ for all $a \in A$. In what follows, we will explicitly write down the Hamiltonian of a lattice model with an anomalous 1-form symmetry $A$.

To define the model, we choose $f$ and $g$ in eq. (3.6) to be the sum of all group elements $a \in A$, i.e., we have $f = g = \bigoplus_{a \in A} a$. For this choice of $f$ and $g$, we can take the dynamical variables on different edges independently. Therefore, the state space $\mathcal{H}$ of the model is given by the tensor product of the local Hilbert spaces on all edges: $\mathcal{H} = \bigotimes_{\text{edges}} \mathbb{C}^{|A|}$. The Hamiltonian is of the form $H = -\sum \hat{T}'_p$, where the local Hamiltonian $\hat{T}'_p$ on a plaquette $p$ is generally given by eq. (4.36).

As an example, we consider the Hamiltonian that consists only of the following three terms:

$$
\hat{T}'_p = \sum_{a \in A} J_x(a)
\begin{array}{c}\vcenter{\hbox{\includegraphics{}}}\end{array}
+ J_y(a)
\begin{array}{c}\vcenter{\hbox{\includegraphics{}}}\end{array}
+ J_z(a)
\begin{array}{c}\vcenter{\hbox{\includegraphics{}}}\end{array}
. \tag{5.2}
$$

Because of the group-like fusion rules, the labels on the orange edges in the above equation are uniquely determined by the configuration of dynamical variables on the honeycomb lattice. A more general Hamiltonian can be obtained by adding the terms given by the products of operators appearing on the right-hand side. For simplicity, we will focus on the Hamiltonian (5.2) in the rest of this subsection.

Let us express the above Hamiltonian in terms of $F$-symbols and $R$-symbols. To this end, we first resolve the 4-valent vertices in eq. (5.2) into trivalent vertices as follows:

$$
\begin{array}{c}\vcenter{\hbox{\includegraphics{}}}\end{array}
=
\begin{array}{c}\vcenter{\hbox{\includegraphics{}}}\end{array}
. \tag{5.3}
$$

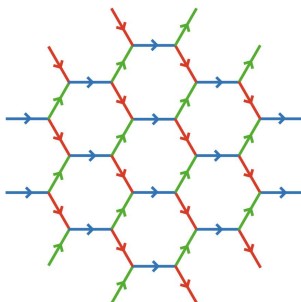

Figure 28: A honeycomb lattice has three types of edges that we call $x$-links, $y$-links, and $z$-links. These edges are written in red, green, and blue in the above figure.

The labels $\Gamma_{ijk}$ on the dotted edges are uniquely determined by the fusion rule $\Gamma_{ijk} = \Gamma_{jk}\Gamma_{ij}$. The above resolution of 4-valent vertices enables us to compute each term in eq. (5.2) as

$$
\cdots = \frac{R(\Gamma_{12},a)F(\Gamma_{26},\Gamma_{26}^{-1}\Gamma_{246}a^{-1},a)F(\Gamma_{24}a^{-1},\Gamma_{12},a)}{F(\Gamma_{46},\Gamma_{24}a^{-1},a)F(\Gamma_{24}a^{-1},a,\Gamma_{12})F(\Gamma_{14},\Gamma_{14}^{-1}\Gamma_{124}a^{-1},a)}\cdots ,
$$

$$
\cdots = \frac{R(\Gamma_{24},a)F(\Gamma_{46},\Gamma_{24},a)F(\Gamma_{48},\Gamma_{48}^{-1}\Gamma_{468},a)}{F(\Gamma_{26},\Gamma_{26}^{-1}\Gamma_{246},a)F(\Gamma_{46},a,\Gamma_{24})F(\Gamma_{68},\Gamma_{46},a)}\cdots ,
$$

$$
\cdots = \frac{F(\Gamma_{48},a,a^{-1}\Gamma_{48}^{-1}\Gamma_{478})}{F(\Gamma_{48},a,a^{-1}\Gamma_{48}^{-1}\Gamma_{468})}\cdots .
$$

If we write the above operators as $\hat{\mathcal{O}}_x(a)$, $\hat{\mathcal{O}}_y(a)$, and $\hat{\mathcal{O}}_z(a)$ respectively, the total Hamiltonian of the model can be written as

$$
H = -\sum_{a\in A}\left[\sum_{x\text{-links}}J_x(a)\hat{\mathcal{O}}_x(a)+\sum_{y\text{-links}}J_y(a)\hat{\mathcal{O}}_y(a)+\sum_{z\text{-links}}J_z(a)\hat{\mathcal{O}}_z(a)\right], \qquad (5.4)
$$

where $x$-links, $y$-links, and $z$-links are depicted by red, green, and blue edges in Figure 28.

**Anomalous $\mathbb{Z}_2$ 1-form symmetry.** As the simplest example, we consider the case of an anomalous $\mathbb{Z}_2$ 1-form symmetry. There are four possible anomalies of a $\mathbb{Z}_2$ 1-form symmetry, which are specified by the following $F$-symbols and $R$-symbols:

$$
(F(\eta,\eta,\eta),R(\eta,\eta)) = (1,1),(1,-1),(-1,i),(-1,-i). \qquad (5.5)
$$

Here, $\eta$ denotes the generator of $\mathbb{Z}_2$. The other components of the $F$-symbols and $R$-symbols are trivial. The $\mathbb{Z}_2$ 1-form symmetries with the above anomalies are called bosonic, fermionic, semionic, and anti-semionic $\mathbb{Z}_2$ 1-form symmetries respectively.[52] For each of these $\mathbb{Z}_2$ 1-form symmetries, we can write down the operators $\hat{\mathcal{O}}_x(\eta)$, $\hat{\mathcal{O}}_y(\eta)$, and $\hat{\mathcal{O}}_z(\eta)$ in terms of the Pauli operators.

---

[52] The bosonic $\mathbb{Z}_2$ 1-form symmetry is non-anomalous, whereas the others are anomalous.

For a bosonic $\mathbb{Z}_2$ 1-form symmetry, the operators $\hat{\mathcal{O}}_x(\eta)$, $\hat{\mathcal{O}}_y(\eta)$, and $\hat{\mathcal{O}}_z(\eta)$ are all given by the Pauli $X$ operator. Thus, the Hamiltonian (5.4) can be written graphically as

$$H = -\sum_{x\text{-links}} J_x \;\;\text{[graphical term]}\;\; - \sum_{y\text{-links}} J_y \;\;\text{[graphical term]}\;\; - \sum_{z\text{-links}} J_z \;\;\text{[graphical term]}, \tag{5.6}$$

where the parameters $J_x := J_x(\eta)$, $J_y := J_y(\eta)$, and $J_z := J_z(\eta)$ are real numbers so that the above Hamiltonian satisfies the Hermiticity condition (4.12). Here and in the rest of this subsection, we set $J_x(1)$, $J_y(1)$, and $J_z(1)$ to zero for the unit element $1 \in \mathbb{Z}_2$ without loss of generality. The above model realizes a trivial phase with bosonic $\mathbb{Z}_2$ 1-form symmetry.

For a fermionic $\mathbb{Z}_2$ 1-form symmetry, the Hamiltonian (5.4) can be written as

$$H = -\sum_{x\text{-links}} J_x \;\;\text{[graphical term]}\;\; - \sum_{y\text{-links}} J_y \;\;\text{[graphical term]}\;\; - \sum_{z\text{-links}} J_z \;\;\text{[graphical term]}. \tag{5.7}$$

The parameters $J_x$, $J_y$, and $J_z$ are again chosen to be real due to the Hermiticity condition (4.12). We note that the qubits on the $z$-links are decoupled from those on the $x$-links and $y$-links. In particular, the qubits on the $z$-links have a uniquely gapped ground state given by a trivial product state. Therefore, in the low-energy limit, the above Hamiltonian reduces to the stacking of decoupled 1+1d quantum spin chains consisting of the qubits on the $x$-links and $y$-links. These quantum spin chains can be coupled to the qubits on the $z$-links by adding the terms such as $\hat{\mathcal{O}}_x(\eta)\hat{\mathcal{O}}_z(\eta)$ and $\hat{\mathcal{O}}_y(\eta)\hat{\mathcal{O}}_z(\eta)$.

For a semionic $\mathbb{Z}_2$ 1-form symmetry, the Hamiltonian (5.4) can be written as

$$H = -\sum_{x\text{-links}} J_x \;\;\text{[graphical term]}\;\; - \sum_{y\text{-links}} J_y \;\;\text{[graphical term]}\;\; - \sum_{z\text{-links}} J_z \;\;\text{[graphical term]}, \tag{5.8}$$

where $CZ$ denotes the controlled-$Z$ operator that acts on the qubits on the two edges connected by a small arc. The Pauli $X$ operator in each term acts on the middle edge after the sequence of the controlled-$Z$ and the Pauli $Z$ operators. We note that the Hamiltonian (5.8) is not Hermitian when the Hermiticity condition (4.12) is satisfied, namely, when $J_x$, $J_y$, and $J_z$ are real numbers. This indicates that the Hermiticity condition (4.12) is invalid for a semionic $\mathbb{Z}_2$ 1-form symmetry. This is because the generator $\eta$ of a semionic $\mathbb{Z}_2$ 1-form symmetry has a non-trivial Frobenius-Schur indicator, which violates the assumption used in the derivation of the Hermiticity condition (4.12). We can make the Hamiltonian (5.8) Hermitian by taking a linear combination of $H$ and its complex conjugate $H^\dagger$. Adding $H^\dagger$ to the original Hamiltonian $H$ does not break the semionic $\mathbb{Z}_2$ 1-form symmetry because $H^\dagger$ commutes with the symmetry action when $H$ does. Similar arguments apply to the case of an anti-semionic $\mathbb{Z}_2$ 1-form symmetry.

## 5.2 Kitaev honeycomb model without a magnetic field

The Kitaev honeycomb model without a magnetic field is an exactly solvable model of qubits on a honeycomb lattice, which exhibits an abelian topological order or a gapless excitation depending on the parameter of the model [166]. As we will see below, this model can be obtained as a variant of the 2+1d fusion surface model.

The fusion 2-category that we use as an input is the 2-category Mod(Ising) of Ising-module categories, where Ising denotes the modular tensor category describing the Ising TQFT. The Ising category consists of three simple objects $\{1, \eta, \sigma\}$, which are subject to the following fusion rules:

$$\eta \otimes \eta \cong 1, \quad \eta \otimes \sigma \cong \sigma \otimes \eta \cong \sigma, \quad \sigma \otimes \sigma \cong 1 \oplus \eta. \tag{5.9}$$

The non-trivial $F$-symbols and $R$-symbols are summarized as follows [166]:

$$(F_\eta^{\sigma\eta\sigma})_{\sigma\sigma} = (F_\sigma^{\eta\sigma\eta})_{\sigma\sigma} = -1, \quad (F_\sigma^{\sigma\sigma\sigma})_{11} = (F_\sigma^{\sigma\sigma\sigma})_{1\eta} = (F_\sigma^{\sigma\sigma\sigma})_{\eta 1} = \frac{1}{\sqrt{2}}, \quad (F_\sigma^{\sigma\sigma\sigma})_{\eta\eta} = -\frac{1}{\sqrt{2}},$$

$$R_1^{\eta\eta} = -1, \quad R_\sigma^{\eta\sigma} = R_\sigma^{\sigma\eta} = -i, \quad R_1^{\sigma\sigma} = e^{-i\pi/8}, \quad R_\eta^{\sigma\sigma} = e^{3i\pi/8}. \tag{5.10}$$

As discussed in full generality in Section 4.4.2, 1-endomorphisms of the unit object of the fusion 2-category Mod(Ising) form the Ising category. In particular, simple 1-morphisms of Mod(Ising) are labeled by simple objects 1, $\eta$, and $\sigma$ of Ising. We note that the invertible object $\eta$ generates an anomalous $\mathbb{Z}_2$ 1-form symmetry.

In order to obtain the Kitaev honeycomb model as a variant of the fusion surface model, we choose both $f$ and $g$ in eq. (3.6) to be $\sigma$. Furthermore, we fix the labels on the edges of the honeycomb lattice to $\sigma$, or in other words, we only consider the sector where all edges are labeled by $\sigma$. We note that the restriction to this sector violates the Ising 1-form symmetry, but still preserves the anomalous $\mathbb{Z}_2$ 1-form symmetry generated by $\eta$. The dynamical variables in this sector are living only on the vertices of the honeycomb lattice. The local Hilbert space on each vertex is given by $\text{Hom}(\sigma \otimes \sigma, \sigma \otimes \sigma) \cong \mathbb{C}^2$, which means that we have a qubit on each vertex. The total Hilbert space of the model is given by the tensor product of the local Hilbert spaces on all vertices.

We define the local Hamiltonian on each plaquette as

$$\tag{5.11}$$

where the green edges are all labeled by $\sigma$. Coupling constants $J_x$, $J_y$, and $J_z$ have to be real due to the Hermiticity condition (4.12). We note that the above Hamiltonian is an example of the local Hamiltonian (4.36) except that the labels on the edges in eq. (5.11) are fixed, whereas those in eq. (4.36) are dynamical. For computational purposes, we resolve each 4-valent vertex into two trivalent vertices as follows:

$$\tag{5.12}$$

Qubits on the left-hand side are living on the 4-valent vertices, whereas qubits on the right-hand side are living on the black edges. The qubits on the right-hand side are denoted by the same letters as the qubits on the left-hand side. After the resolution of the 4-valent

vertices, we can evaluate each term on the right-hand side of eq. (5.11) as

In terms of the Pauli operators, we can write the above operators as $-Y_{124}X_{246}$, $Y_{246}X_{468}$, and $Z_{468}Z_{478}$ respectively, where $X_{ijk}$, $Y_{ijk}$, and $Z_{ijk}$ denote the Pauli $X$, $Y$, and $Z$ operators acting on the qubit $\Gamma_{ijk}$. If we rotate the bases of qubits $\Gamma_{124}$ and $\Gamma_{468}$ by $\pi/4$, the first and the second operators become $X_{124}X_{246}$ and $Y_{246}Y_{468}$ respectively. Therefore, the Hamiltonian in the rotated basis can be written as

$$H = -\sum_{x\text{-links}} J_x X_i X_j - \sum_{y\text{-links}} J_y Y_i Y_j - \sum_{z\text{-links}} J_z Z_i Z_j, \tag{5.13}$$

where $x$-links, $y$-links, and $z$-links are three different types of edges shown in Figure 28. The above is the Hamiltonian of the Kitaev honeycomb model without a magnetic field [166]. Remarkably, our formulation makes the anomalous $\mathbb{Z}_2$ 1-form symmetry of the Kitaev honeycomb model manifest. The symmetry operator on a closed loop indeed agrees with the loop operator defined in Kitaev's original paper [166]. This symmetry guarantees that the Kitaev honeycomb model without a magnetic field realizes non-trivial phases everywhere in the phase diagram. We note that applying a magnetic field explicitly breaks the anomalous $\mathbb{Z}_2$ 1-form symmetry.

## 5.3 Non-chiral topological phases with fusion 2-category symmetries

In this subsection, we will sketch out how to obtain the 2+1d fusion surface models that realize non-chiral topological phases with fusion 2-category symmetries. As we discussed in Section 4, the fusion surface model is the quantum counterpart of the 3d height model, which is obtained by putting the 4d Douglas-Reutter TFT on a slab as shown in Figure 18. The dynamics of the fusion surface model is determined by the choice of a decorated boundary condition on the right boundary of the slab. In particular, when the decorated boundary is topological, the corresponding fusion surface model realizes a topological phase with fusion 2-category symmetry.

**Dirichlet boundary and spontaneous symmetry breaking.** The simplest example of a topological boundary condition is the Dirichlet boundary condition, which is defined by the trivial coloring on the decorated boundary. Specifically, for the Dirichlet boundary condition, the simple objects $\rho, \sigma, \lambda$, and the simple 1-morphisms $f, g$ in eq. (3.6) are the unit object $I$ and the identity 1-morphism $\mathbf{1}_I$ respectively. This implies that the graphical representation (4.13) of a state is given by a planar fusion diagram on a honeycomb lattice. We note that all the plaquettes of the honeycomb lattice are labeled by the same simple object because the simple objects on the adjacent plaquettes have to be connected. Therefore, the state space $\mathcal{H}_0$ splits into sectors labeled by (representatives

of connected components of) simple objects of a fusion 2-category $\mathcal{C}$, namely, we have $\mathcal{H}_0 = \bigoplus_{X \in \pi_0 \mathcal{C}} \mathcal{H}_0^X$. Each sector $\mathcal{H}_0^X$ is the image of the projector $\prod_p \hat{B}_p$, where $\hat{B}_p$ is the plaquette operator defined by

$$\hat{B}_p \quad \parbox{3cm}{\centering [diagram]} \quad = \sum_{x \in \mathrm{End}(X)} \frac{\|x\|}{\dim(\mathrm{End}(X))} \quad \parbox{3cm}{\centering [diagram]} . \qquad (5.14)$$

Here, $\|x\| = \dim(x)/\dim(X)$ is the norm of a simple 1-morphism $x$, i.e., the quantum dimension of $x$ viewed as a simple object of a fusion 1-category $\mathrm{End}(X)$. The Hamiltonian (4.20) acting on the state space $\mathcal{H}_0$ is (proportional to) the identity operator because the weight $A(\mathcal{F}_{\mathrm{int}})$ is zero when the coloring $\mathcal{F}_{\mathrm{int}}$ is non-trivial. Thus, the ground state subspace of the model is $\mathcal{H}_0$ itself. Since the plaquette operator (5.14) is the same as that of the Levin-Wen model for an input fusion 1-category $\mathrm{End}(X)$ [125], each sector $\mathcal{H}_0^X$ of our model realizes a non-chiral topological order described by the Drinfeld center $Z(\mathrm{End}(X))$ of $\mathrm{End}(X)$.[53] The non-chiral topological orders realized on different sectors are mixed by the action of a fusion 2-category symmetry $\mathcal{C}$. Physically, this means that the fusion 2-category symmetry $\mathcal{C}$ is spontaneously broken.

We note that the projection to $\mathcal{H}_0$ can also be implemented dynamically by the Hamiltonian $H = -\sum_p \hat{B}_p$ acting on a larger state space $\mathcal{H}$ spanned by all possible fusion diagrams on the honeycomb lattice. However, in this case, the fusion 2-category symmetry is not exact on the lattice but emergent in the low-energy limit.

**General topological boundaries and non-chiral topological phases.** A more general topological boundary condition gives rise to a more general topological phase with fusion 2-category symmetry. In general, topological boundaries of the Douglas-Reutter TFT $\mathrm{DR}(\mathcal{C})$ that give rise to non-chiral topological phases with $\mathcal{C}$ symmetry are expected to be in one-to-one correspondence with (the equivalence classes of) finite semisimple module 2-categories over $\mathcal{C}$.[54] Since (the equivalence classes of) finite semisimple module 2-categories over $\mathcal{C}$ are in one-to-one correspondence with (the Morita equivalence classes of) separable algebras in $\mathcal{C}$ [169],[55] there should also be a one-to-one correspondence between non-chiral topological boundaries of $\mathrm{DR}(\mathcal{C})$ and separable algebras in $\mathcal{C}$. In particular, the Dirichlet boundary corresponds to the trivial algebra $I \in \mathcal{C}$. A general non-chiral topological boundary would be realized by condensing a separable algebra $A \in \mathcal{C}$ on the Dirichlet boundary. In other words, the coloring $\mathcal{F}$ on a non-chiral topological boundary would be given by the condensation of a separable algebra $A$, which is a fine mesh of topological surfaces labeled by $A$ [56, 171, 172]. Indeed, as we will see below, when the coloring $\mathcal{F}$ is the condensation of a separable algebra $A \in \mathcal{C}$, the corresponding 2+1d fusion surface model has a commuting projector Hamiltonian, which suggests that the model realizes a non-chiral topological phase with fusion 2-category symmetry $\mathcal{C}$.

In order to obtain the 2+1d fusion surface models for general non-chiral topological phases, we first briefly recall the definition of a separable algebra in a fusion 2-category $\mathcal{C}$ [173]. An algebra $A$ in $\mathcal{C}$ is an object equipped with a multiplication 1-morphism

---

[53]The constraints from the vertex terms of the Levin-Wen model are already imposed on the state space.

[54]This is a 4d analogue of the fact that topological boundaries of 3d Turaev-Viro TFT are in one-to-one correspondence with (the equivalence classes of) finite semisimple module categories over the input fusion 1-category [19, 122, 123, 142, 143]. For the 4d Dijkgraaf-Witten theory, the correspondence between topological boundaries and module 2-categories over $2\mathrm{Vec}_G$ is studied in, e.g., [167, 168].

[55]This is a categorified version of Ostrik's theorem [170].

$m : A\square A \to A$ that is associative up to coherence 2-isomorphism $\mu$ satisfying[56]

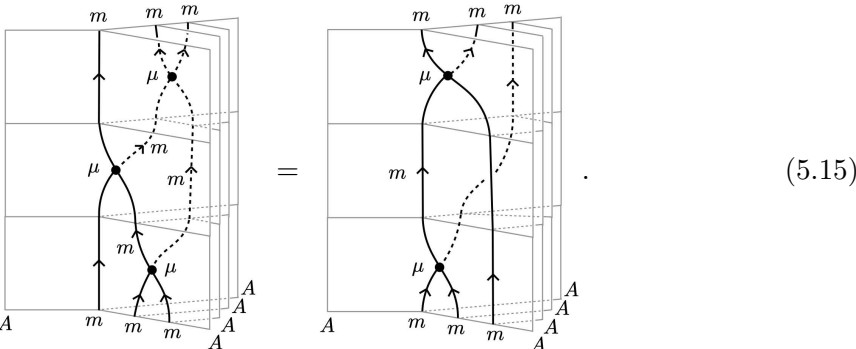

$$\tag{5.15}$$

An algebra $A \in \mathcal{C}$ is called a rigid algebra if the multiplication 1-morphism $m : A\square A \to A$ has a dual 1-morphism $m^* : A \to A\square A$. There are a lot of coherence conditions associated with this duality, which we assume implicitly in the following, see [173,174] for the precise definition. A rigid algebra $A \in \mathcal{C}$ is called a separable algebra if the multiplication 1-morphism $m$ and its dual $m^*$ satisfy the following conditions:

$$\tag{5.16}$$

$$\tag{5.17}$$

The 2-morphisms in eq. (5.17) should be regarded as appropriate duals of the associativity 2-isomorphism $\mu$ and its inverse $\mu^{-1}$. By a slight abuse of notation, we write these 2-isomorphisms simply as $\mu$ and $\mu^{-1}$ in the above equation.[57] It is conjectured in [175] and is proven in [176] that rigid algebras in a fusion 2-category over $\mathbb{C}$ are automatically separable. We note that separable algebras are closely related to the orbifold data of 3d topological field theories [171,172].

Based on the above definition of a separable algebra, we now write down the Hamiltonians of the fusion surface models that realize non-chiral topological phases with fusion 2-category symmetry $\mathcal{C}$. As we mentioned above, the fusion surface model for a non-chiral topological phase is obtained by choosing the coloring $\mathcal{F}$ to be the condensation of a separable algebra $A \in \mathcal{C}$, i.e., $\mathcal{F}_{ij} = A, \mathcal{F}_{ijk} = m$, and $\mathcal{F}_{ijkl} = \mu$. The Hamiltonian of this model is given by $H = -\sum_p \hat{T}_p$, where the local Hamiltonian $\hat{T}_p$ is represented by the following fusion diagram:

$$\tag{5.18}$$

---

[56]Precisely, an algebra in a fusion 2-category is also equipped with a 1-morphism $i : I \to A$ that satisfies the unitality condition up to coherence 2-isomorphism. We will not use this datum explicitly in the following discussions.

[57]These 2-isomorphisms are denoted by $\psi^l$ and $\psi^r$ in [173].

Here, we suppose that the state space of the model is $\mathcal{H}_0$ rather than $\mathcal{H}$ so that the model has an exact fusion 2-category symmetry. It is obvious that the plaquette term $\hat{T}_p$ commute with another plaquette term $\hat{T}_{p'}$ when the plaquettes $p$ and $p'$ are apart from each other. The commutativity of $\hat{T}_p$ and $\hat{T}_{p'}$ for adjacent plaquettes also follows from eqs. (5.15) and (5.17). For example, we have

$$\hat{T}_p \hat{T}_{p'} = \quad = \quad = \quad = \hat{T}_{p'} \hat{T}_p. \qquad (5.19)$$

Furthermore, the plaquette term $\hat{T}_p$ is a projector, which means that it satisfies

$$\hat{T}_p^2 = \quad = \quad = \hat{T}_p. \qquad (5.20)$$

Here, we used eqs. (5.15) and (5.16) in the first and the third equalities respectively. The second equality follows from the identity $\mu^{-1}\mu = \mu\mu^{-1} = \text{id}$. Thus, we find that the 2+1d fusion surface model obtained from a separable algebra $A \in \mathcal{C}$ has a commuting projector Hamiltonian, which strongly suggests that this model realizes a non-chiral topological phase with fusion 2-category symmetry $\mathcal{C}$.

Let us finally consider some simple examples. When $\mathcal{C}$ is the 2-category 2Vec of finite semisimple 1-categories, separable algebras are given by multifusion 1-categories [173]. In this case, we expect that eq. (5.18) reduces to the Hamiltonian of the Levin-Wen model, which realizes the most general non-chiral topological order without symmetry.[58] More generally, when $\mathcal{C}$ is the 2-category 2Vec$_G$ of $G$-graded finite semisimple 1-categories, separable algebras are given by $G$-graded multifusion 1-categories [173]. In this case, we expect that eq. (5.18) reduces to the Hamiltonian of the symmetry enriched Levin-Wen model [178, 179], which realizes the most general non-chiral topological order enriched by a finite group symmetry $G$ [3].

## Acknowledgments

KI is supported by FoPM, WINGS Program, the University of Tokyo, and also by JSPS Research Fellowship for Young Scientists. KO is supported by JSPS KAKENHI Grant-in-Aid No.22K13969 and the Simons Collaboration on Global Categorical Symmetries.

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
