# Peer review of "Fusion Surface Models: 2+1d Lattice Models from Fusion 2-Categories"

_SciPost Physics Core_

## Round 1 · Referee Report · Anonymous · 2024-3-3

Report

Regarding item 6 of Report 1, it is not true in general that a "Δ-separable Frobenius algebra in a pivotal fusion category is automatically symmetric". For example, the cited reference by Fuchs-Runkel-Schweigert proves symmetry under the additional assumptions "sovereign" and "haploid". The symmetry condition can be thought of as a compatibility condition between the ambient structure of and on adjoints (here: pivotality) and the underlying condensation monad. Such compatibilities also appear in higher dimensions and constitute the main difference between condensation monads and orbifold data.

---

## Round 1 · Referee Report · Anonymous · 2024-3-21

Report

The suggestions, comments and questions have been addressed in the revised version.

---

## Round 1 · Referee Report · Anonymous · 2024-4-17

Report

I would like to thank the authors for their replies to my questions and for taking into account the points raised.

Recommendation

Publish (surpasses expectations and criteria for this Journal; among top 10%)

---

## Round 1 · List of Changes

Please see the replies below for a detailed list of changes.

---

## Editorial Decision

in_voting